# MOF-in-COF molecular sieving membrane for selective hydrogen separation

Hongwei Fan[1], Manhua Peng[2], Ina Strauss[1], Alexander Mundstock[1], Hong Meng[3 ✉] & Jürgen Caro [1,4 ✉]

Covalent organic frameworks (COFs) are promising materials for advanced molecular-separation membranes, but their wide nanometer-sized pores prevent selective gas separation through molecular sieving. Herein, we propose a MOF-in-COF concept for the confined growth of metal-organic framework (MOFs) inside a supported COF layer to pre-pare MOF-in-COF membranes. These membranes feature a unique MOF-in-COF micro/nanopore network, presumably due to the formation of MOFs as a pearl string-like chain of unit cells in the 1D channel of 2D COFs. The MOF-in-COF membranes exhibit an excellent hydrogen permeance (>3000 GPU) together with a significant enhancement of separation selectivity of hydrogen over other gases. The superior separation performance for $H_2/CO_2$ and $H_2/CH_4$ surpasses the Robeson upper bounds, benefiting from the synergy combining precise size sieving and fast molecular transport through the MOF-in-COF channels. The synthesis of different combinations of MOFs and COFs in robust MOF-in-COF membranes demonstrates the versatility of our design strategy.

[1] Institute of Physical Chemistry and Electrochemistry, Leibniz Universität Hannover, Callinstraße 3A, 30167 Hannover, Germany. [2] Institut für Festkörperphysik, Leibniz Universität Hannover, Appelstrasse 2, 30167 Hannover, Germany. [3] Beijing Key Laboratory of Membrane Science and Technology, Beijing University of Chemical Technology, 100029 Beijing, PR China. [4] School of Chemistry and Chemical Engineering, South China University of Technology, 510640 Guangzhou, PR China. ✉email: menghong@mail.buct.edu.cn; juergen.caro@pci.uni-hannover.de

Membrane-based gas-separation processes have created a high interest in petrochemical industry due to their high efficiency, low energy costs and convenient operation[1,2]. Conventional polymeric membranes often suffer from a "trade-off" between gas permeability and selectivity, and therefore, the development of novel membrane materials with adequate performance is a challenging task to meet the separation requirements under practical process conditions[3,4]. Molecular sieving membranes with abundant and uniform pore structures that can break the Robeson limit are desirable for energy-efficient gas separation[4]. To this end, typical porous materials, such as zeolites[5–7], metal–organic frameworks (MOFs)[8–13], microporous organic materials[14–18], or two-dimensional (2D) layered materials[19–23] as sieving membranes have been extensively investigated over the past decade. Covalent organic frameworks (COFs) are an emerging new class of crystalline porous materials[24,25], which are formed by atomically precise integration of organic units via strong covalent bonds. The COFs, especially the Schiff-based COF family, not only possess inherent properties like high porosity, versatile and tunable pore size, well-defined pore structure and readily tailored functionalities, but have superior thermochemical stability in comparison with the coordination-based MOFs[26,27]. These fascinating features make COFs excellent candidates for constructing new-generation molecular sieving membranes for advanced separation[28–33]. Nevertheless, there is very limited progress regarding COF membranes in selective gas separation so far[34–36]. In addition to the difficulties related to the fabrication of defect-free and continuous COF selective layers, the main reason is due to the wide nanometer-sized pores of the COF family (typically 0.8–5 nm)[37,38], which are much larger than the kinetic diameter of common gas molecules (0.25–0.50 nm)[39]. The reported approaches to mitigate this including the introduction of side groups into COF cavity walls[40–44] or staggered stacking of 2D COFs[45–49] can reduce the aperture size into the microporous range, but it is difficult to form effective and uniform molecular channels in the resulting membrane for precise size sieving in gas mixture separation.

Constructing multi-component hierarchical porous materials such as the hybrids between MOFs and COFs[50] provides a new route for the development of COF-based membranes suited for gas separation with the synergy of microporous MOFs and nanoporous COFs. As dual-layer membranes, two representatives of [COF-300]-[Zn$_2$(bdc)$_2$(dabco)] and [COF-300]-[ZIF-8] membranes reported by Fu and co-workers[51], show enhanced separation selectivity (~13) for H$_2$/CO$_2$ gas mixtures compared with the individual COF and MOF membranes (~9). Another [COF-300]-[UiO-66] membrane reported by Das and co-workers[52] exhibits even higher H$_2$/CO$_2$ selectivity of 17.2. However, the performance of these membranes mainly relies on the interfacial interaction between the MOF and COF layers, but the sieving property of the ordered pore structures in MOFs and COFs have not been brought fully into play, which greatly restricts any further improvement in separation selectivity.

It is well known that the unit cell is the smallest repeating unit that constitutes a crystal structure such as a MOF. Generally, the unit cell size of MOFs falls in a range of several nanometers (such as the cubic cell dimension of ZIF-8 and ZIF-67 is $a = b = c = $ ~1.7 nm)[53,54], which is close to the inner pore width of COFs. This means, theoretically the COF pore can accommodate unit cell-sized MOFs as cage or chain. It is expected that a membrane with such MOF-in-COF pore structures will exhibit excellent sieving performance together with a high molecular transfer rate for the separation of gas mixtures upon the well-defined hierarchical micro/nanochannels. Unfortunately, to date there is no concept to exploit the MOF-in-COF materials as membranes for separation applications.

Herein, we present the fabrication of MOF-in-COF membranes and their use for selective gas separation by using the facile approach of confinement synthesis of MOFs in the continuous 2D COF membrane layers. The resulting membranes have a unique MOF-in-COF micro/nanopore network, exhibiting an excellent performance in terms of ultrahigh H$_2$ permeance and separation selectivity for gas mixtures such as H$_2$/CO$_2$ and H$_2$/CH$_4$. To demonstrate the broad applicability of this design concept, various MOF-in-COF membranes are developed in this study and their performance for selective H$_2$ separation is investigated. The MOF-in-COF strategy in this study may inspire the design of high-performance membrane materials and also promote the advances in gas-separation applications of COF-based membranes.

## Results

**Preparation of MOF-in-COF membrane.** As a proof of concept, we choose the Co-based zeolitic imidazolate framework ZIF-67 and the 2D ketoenamine-linked TpPa-1[55] as building blocks because they belong to the most stable representatives of MOFs and COFs. Moreover, ZIF-67 has an attractive molecular sieve effect for gas mixtures due to its ultramicropore system sized at about 0.34 nm as determined by X-ray diffraction[56]. Also the size of a sodalite (SOD) cage (cubic, space group $I\bar{4}3$ m, $a = 16.9589$ Å) forming as unit cell the ZIF-67 structure is just smaller than the pore size of TpPa-1 (~1.83 nm). Before the preparation of MOF-in-COF membrane, a continuous TpPa-1 layer was grown onto a porous $\alpha$-Al$_2$O$_3$ substrate surface via an in situ solvothermal synthesis method, with the aim to provide the nano-confined template for the subsequent MOF growth. A facile two-stage immersion process is employed to prepare the MOF-in-COF membrane at room temperature, as illustrated in Fig. 1a. First, the supported TpPa-1 layer was vertically immersed into a cobalt nitrate hexahydrate (Co(NO$_3$)$_2$·6H$_2$O) solution for 24 h to adequately adsorb Co$^{2+}$ions (Supplementary Fig. 1). Afterward, a 2-methylimidazole (2-meIM) solution was added to allow the confined growth of ZIF-67 into the TpPa-1 layer. It is expected that one SOD cage as a unit cell of ZIF-67 is formed inside the 1D pore channels of TpPa-1 to give the ZIF-67-in-TpPa-1 membrane. It should be noted that this concept is completely different from other COF-MOF membranes with MOF layers grown on COF layers, or in reversed order[51,52]. The morphologies of the membranes are characterized by SEM. We can see that a continuous TpPa-1 layer with a thickness of about 1 μm (Fig. 1c, f) is grown on the porous alumina substrate (Fig. 1b, e). After the two-stage immersion process, the membrane morphology and thickness (Fig. 1d, g) are almost unchanged, and no ZIF-67 crystals (Supplementary Fig. 2) are formed or deposited on the outer membrane surface. The obvious changes in membrane colors imply that the ZIF-67 might have been grown inside the TpPa-1 layer (Fig. 1h). Energy-dispersive X-ray spectroscopy (EDXS) (Fig. 1i, j) reveals a sharp transition between the ZIF-67-in-TpPa-1 membrane layer (C, Co signals with a combination of red and green) and the alumina substrate (Al signals), indicating that no detectable TpPa-1 or ZIF-67 crystals nucleated into the bulk ceramic substrate. In addition, the uniform Co signals from the surface EDXS (Supplementary Fig. 3) suggest a good dispersion of ZIF-67 in the membrane.

**Structural analysis of MOF-in-COF membrane.** As shown in Fig. 2a, both the TpPa-1 layer and ZIF-67-in-TpPa-1 membrane exhibit a diffraction peak at $2\theta = 4.7°$ corresponding to the (100 facet) reflection planes of TpPa-1. This finding from XRD indicates the successful growth of a TpPa-1 layer on the support, and that no structural damage due to the incorporation of ZIF-67

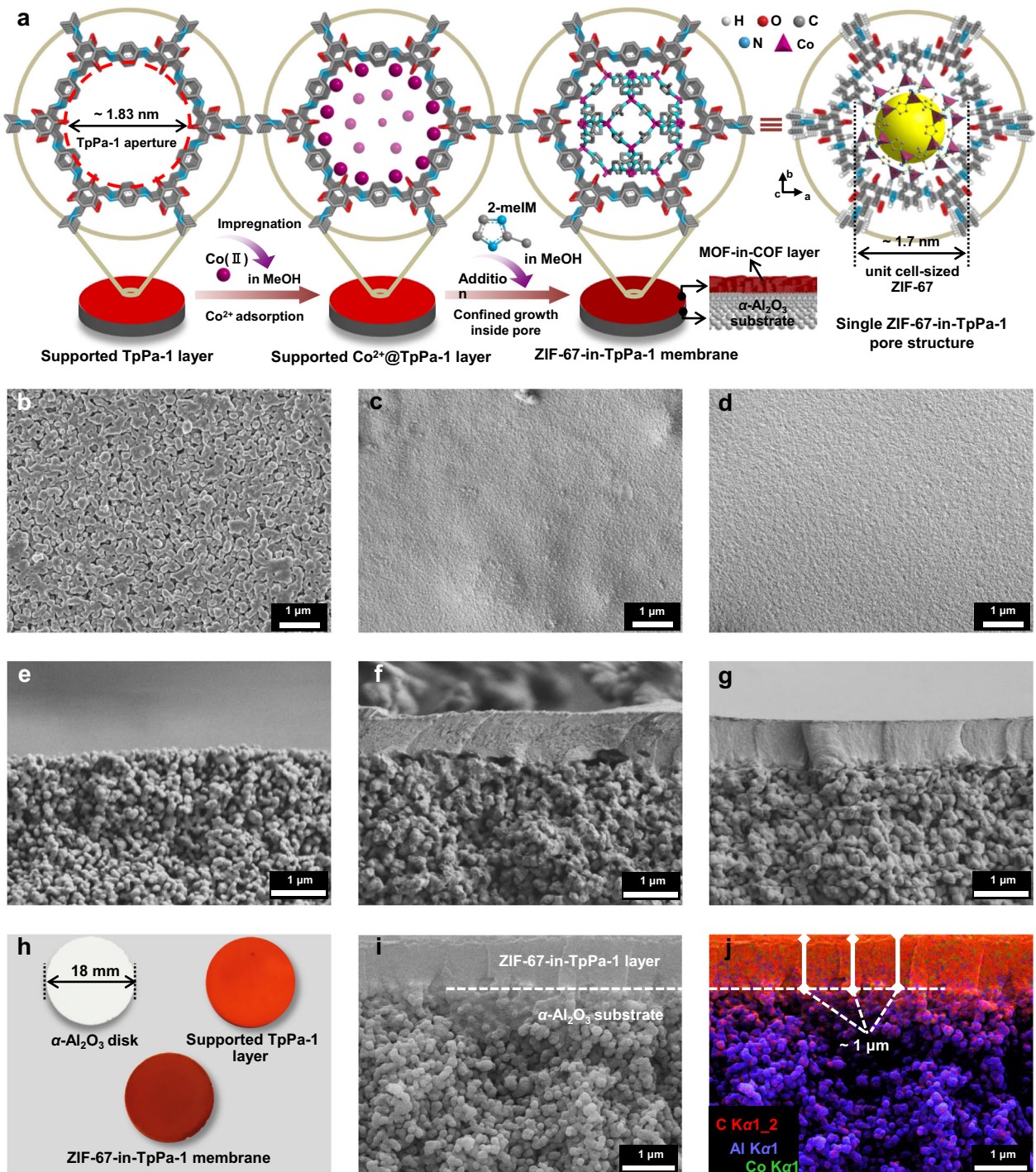

**Fig. 1 Preparation of MOF-in-COF membrane and morphological characterization. a** Scheme depicting ZIF-67-in-TpPa-1 membrane synthesis and schematic of single pore structure. Top-view (**b–d**) and cross-sectional (**e–g**, **i**) SEM images of porous α-Al₂O₃ substrate (**b**, **e**), supported TpPa-1 layer (**c**, **f**) and ZIF-67-in-TpPa-1 membrane (**d**, **g**, **i**). **h** Optical photograph of membranes. **j** EDXS mapping and elemental distributions corresponding to (**i**).

happens. No obvious characteristic diffraction signals of ZIF-67 can be detected for the ZIF-67-in-TpPa-1 membrane, indicating that the ZIF-67 was probably formed in tiny size of a few unit cells inside the TpPa-1 pore, rather than as thin layers covering the outer surface of the TpPa-1 layer. However, measuring the powder XRD of ZIF-67-in-TpPa-1 scraped from the substrate, there can be detected several weak diffraction peaks assigned to the ZIF-67 appearing in the magnified XRD zone (Supplementary Fig. 4). This finding clearly indicates the formation of ZIF-67 inside the COF layer. The diffraction signals of the TpPa-1 layer

are not as strong as those of the powder (Supplementary Figs. 5, 6), probably owing to an oriented growth of 2D TpPa-1 in its eclipsed stacking structure parallel to the substrate surface as reported elsewhere[57]. From FTIR spectra in Fig. 2b, the TpPa-1 layer and ZIF-67-in-TpPa-1 membrane show strong signals at about 1575–1580 cm⁻¹ and 1242–1255 cm⁻¹, arising from the characteristic C = C and C–N stretching of the TpPa-1 matrix with a ketoenamine form. The ZIF-67 displays characteristic peaks at about 1575 cm⁻¹ corresponding to the C = N bond in imidazole ring, and other bands in the range of 600–1500 cm⁻¹

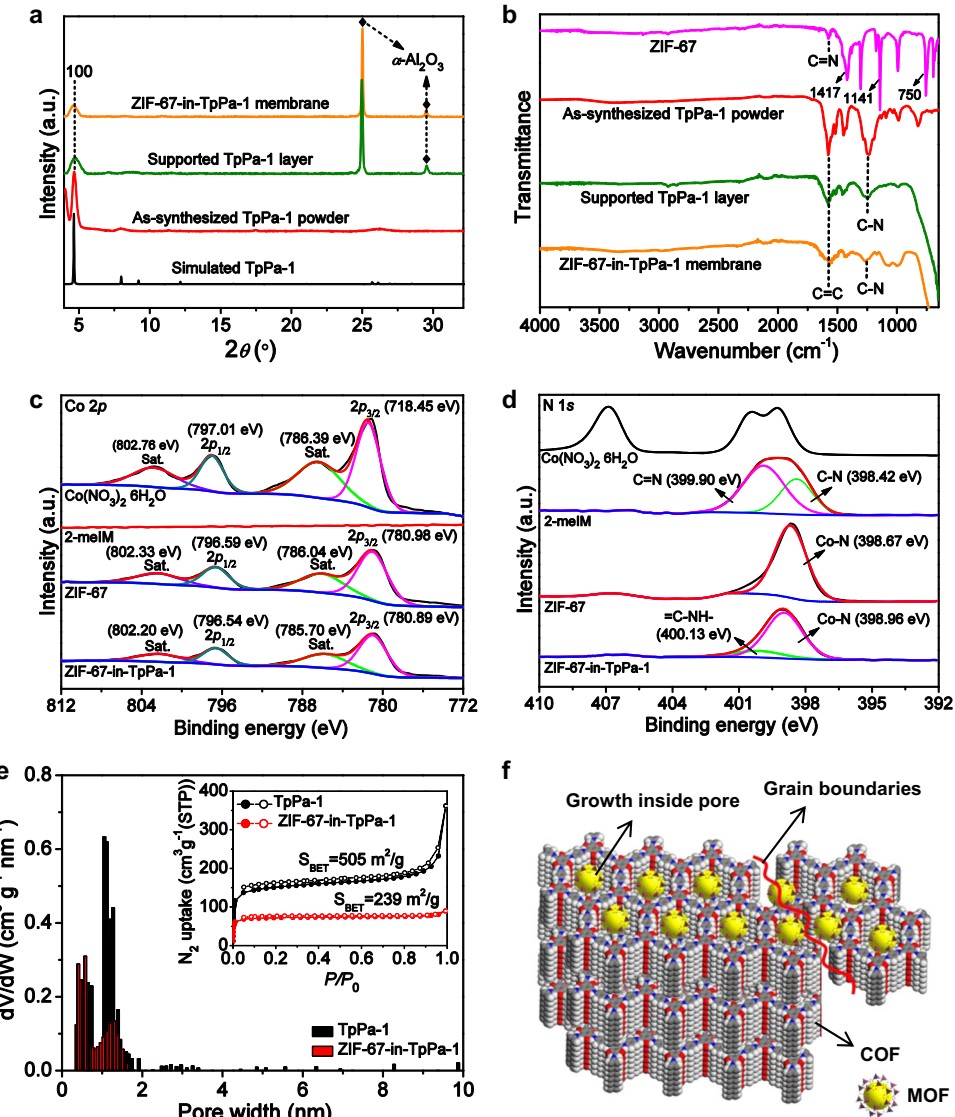

**Fig. 2 Structural analysis. a** XRD patterns. **b** FTIR spectra. **c, d** High-resolution XPS spectra of deconvoluted Co2p and N1s in different samples. **e** Pore-size distribution of powdered TpPa-1 and ZIF-67-in-TpPa-1 with inserted nitrogen adsorption–desorption isotherms measured at 77 K. Adsorption, closed; desorption, open. **f** Schematic illustrating possible growth sites of MOF inside COF layers.

associated with the stretching and bending modes of the imidazole ring. However, it is difficult to detect these signals of ZIF-67 in the ZIF-67-in-TpPa-1 membrane because of the low content and unit cell-sized ZIF-67 with a highly dispersed state in the TpPa-1 layer.

To further detect the ZIF-67 in the membrane, the chemical composition and elemental state were analyzed by X-ray photoelectron spectroscopy (XPS). The full XPS survey spectra (Supplementary Fig. 7) indicate the presence of nitrogen, oxygen, carbon, and cobalt in the ZIF-67-in-TpPa-1 membrane. The high-resolution spectra of the deconvoluted C1s and O1s (Supplementary Fig. 8) display the characteristic energy peaks of the TpPa-1[58]. The high-resolution spectrum of the deconvoluted Co2p and N1s as proof of the formation of the ZIF-67 in different samples are shown in Fig. 2c, d. For the ZIF-67-in-TpPa-1 membrane, two main peaks at 780.89 and 796.54 eV, separated by about 15 eV, are assigned to Co2p$_{3/2}$ and Co2p$_{1/2}$, respectively. Simultaneously, two small and indistinctive peaks are located at 785.70 and 802.20 eV, which are typical Co(II) shakeup satellite (Sat.) peaks of Co(II). The similar electronic state with close binding energies to that of the Co(II) in ZIF-67

(with two main Co2p peaks at 780.98 and 796.59 eV)[59] demonstrates that the cobalt signals in the ZIF-67-in-TpPa-1 membrane come indeed from a Co in a ZIF-67 framework. Moreover, a negative shift in binding energy suggests that there is probably an interaction between Co(II) in ZIF-67 and the N of the COF matrix. In addition, compared to the Co2p spectra in Co(NO$_3$)$_2$·6H$_2$O, a distinct shift in the Co binding energies was clearly observed, which further evidences the coordination of a Co-N interaction in the membrane. Looking at nitrogen, the deconvoluted N1s spectra contain two peaks in the ZIF-67-in-TpPa-1 membrane: 400.13 eV, corresponding to the enamine nitrogen (C = C–NH–) in the TpPa-1, and 398.96 eV, which is associated with the Co-N[60]. The existence of Co-N with the distinct difference in N binding energies from the 2-meIM (with C–N at 398.42 eV and C = N at 399.90 eV) illustrates the complete coordination reaction of 2-meIM and Co$^{2+}$forming ZIF-67 in the hosting TpPa-1 structure.

Apparently, MOF growth inside the TpPa-1 results in a MOF-in-COF pore structure, which can be indirectly proved from the changes in specific surface area and pore-size distribution. Thus, we synthesized corresponding powdered ZIF-67-in-TpPa-1

samples (Supplementary Fig. 9) for measurement of nitrogen adsorption–desorption isotherms. As shown in Fig. 2e, the Brunauer-Emmett-Teller (BET) surface area of ZIF-67-in-TpPa-1 is 239 $m^2 g^{-1}$, and the experimental pore-size distribution (EPSD) is concentrated in the range of 0.29-0.5 nm, both less than the original TpPa-1. The decrease in BET surface area can be attributed to the partial pore blocking and space occupation by the ZIF-67 nucleation, and the lowering of order of degree of TpPa-1 framework caused by exfoliation. The EPSD also illustrates that incorporation of MOFs could effectively narrow the COF pore size in a suitable range for gas-permeation studies. It is worth noting that the experimental pore-size distribution of TpPa-1 is slightly smaller than the intrinsic pore size (~1.83 nm), which is possibly due to the partly staggered stacking of the 2D TpPa-1 layers along the c direction.

Furthermore, we tried TEM to detect ZIF-67 in TpPa-1, but the result is not as expected. There are no visible ZIF-67 lattice fringes and only some dark spots emerged in the TpPa-1 from the high-resolution TEM image (Supplementary Fig. 10). This is probably due to the instability of the unit cell-sized MOF which is damaged under the high energy beam of electrons. In view of the above analysis, we are sure that ZIF-67 species have been formed inside

the TpPa-1 layer. In addition, there exist possibly some defects at the nanoscale (a few nm or even larger) at the COF grain boundaries or in the gaps between the COF layers. Such defective voids provide also sufficient space for ZIF-67 formation. Some MOFs, therefore, might also grow in the COFs grain boundaries (Fig. 2f) thus repairing these defects, which improves gas selectivity.

**Gas-separation performance of MOF-in-COF membrane.** Gas-separation performance was measured following the Wicke–Kallenbach method by placing the MOF-in-COF membrane into a home-made module with $N_2$ as sweep gas (Fig. 3a). Before gas permeation, an on-stream activation was carried out at 393 K to eliminate potential guest molecules inside the pores of the membrane by using an equimolar $H_2/CO_2$ mixture as sweep and feed. Single gases of $H_2$, $CO_2$, $CH_4$, $C_3H_6$, and $C_3H_8$ as well as equimolar binary mixtures of $H_2$ with $CO_2$, $CH_4$, $C_3H_6$, and $C_3H_8$ were tested at room temperature (298 K) and 1 bar, respectively, and the results are summarized in Fig. 3b and Supplementary Table 1. It can be seen that for the ZIF-67-in-TpPa-1 membrane, the $H_2$ permeance of ~3800 GPU is much higher than those of the other gases. The ideal separation factors (ISFs, calculated as the ratio of the single component permeances) of $H_2$ from $CO_2$,

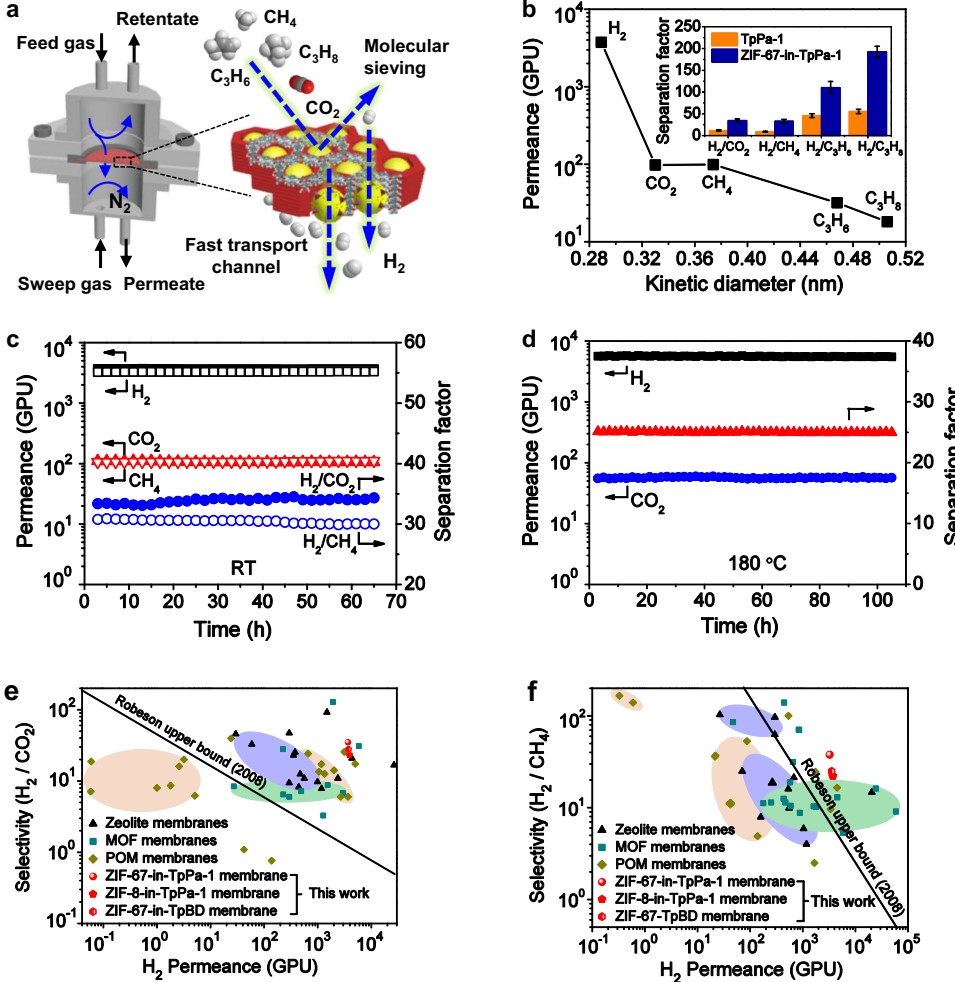

**Fig. 3 Gas-permeation performance. a** Home-made gas-permeation module and schematic illustrating gas transport through ZIF-67-in-TpPa-1 membrane. **b** Single-gas permeances of ZIF-67-in-TpPa-1 membrane as a function of kinetic diameter of permeating molecules at 298 K and 1 bar. The inset shows the mixed-gas-separation factor of ZIF-67-in-TpPa-1 membrane and TpPa-1 membrane for $H_2$ over other gases. Long-term tests of ZIF-67-in-TpPa-1 membrane for (**c**) equimolar gas mixture of $H_2/CO_2$ and $H_2/CH_4$ at 298 K and 1 bar, and (**d**) equimolar gas mixture of $H_2/CO_2$ at 453 K (180 °C) and 1 bar. Mixed-gas selectivities of (**e**) $H_2/CO_2$ and (**f**) $H_2/CH_4$, as a function of $H_2$ permeance for our MOF-in-COF membranes compared with literature data. Information on the data points is given in Supplementary Tables 2, 3.

$CH_4$, $C_3H_6$, and $C_3H_8$ are 38.3, 37.8, 117.7, and 206.8, considerably exceeding the corresponding Knudsen constants (4.7, 2.8, 4.6, 4.7). This demonstrates the molecular sieving property of the ZIF-67-in-TpPa-1 membrane, which is expected to display high performance for selective $H_2$ separation in mixed-gas permeation. As shown in Fig. 3b, the real mixed-gas-separation factors (SFs) of the ZIF-67-in-TpPa-1 membrane for equimolar $H_2/CO_2$, $H_2/CH_4$, $H_2/C_3H_6$, and $C_3H_8$ gas pairs can reach 34.9, 33.3, 110.5, and 192.7, respectively. There is a significant enhancement in separation selectivity as compared to that of the TpPa-1 layer without ZIF-67 (Supplementary Fig. 11), and simultaneously the ZIF-67-in-TpPa-1 membrane keeps high $H_2$ permeance (>3000 GPU). Moreover, the performance is also far better than that of the pure ZIF-67 membrane (Supplementary Figs. 12, 13) in terms of both selectivity and $H_2$ permeance in the separation of equimolar $H_2/CO_2$ and $H_2/CH_4$ gas pairs (Supplementary Fig. 14).

The excellent performance can be explained by the formation of a MOF-in-COF micro/nanopore network in the selective layer. In the confined nanoscopic space, the encapsulated MOF could have a smaller effective pore size than that of the bulk MOF, probably as a result of the increased rigidity of the confined MOF lattice[61,62]. In this case, it could endow more precise molecular sieving channels for the penetrating molecules, which results in the significantly enhanced separation selectivity of $H_2$ over other more bulky gases. This concept could be further proved by the comparison between predicted permeation results using Maxwell model[63] and our experimental data. The calculated $P(H_2)_{MOF-in-COF}$ of 4014 GPU is higher than our experimental permeance of 3252 GPU, while the predicted $H_2/CH_4$ selectivity is about 9.2, much is much lower than the measured value of 33.3. The disagreement between them indicates that the formed MOFs are not simply dispersed in the COF matrix, but a unique MOF-in-COF pore structure with good interfacial interaction between the MOFs and COFs. Moreover, the vertical 1D channel of 2D COFs unlike the zigzag-type pores has smaller flow resistance, which enables the ultrafast transfer of $H_2$ molecules through the selective layer. In other words, there is a synergy in the performance enhancement of the MOF-in-COF membrane. In addition, it cannot be excluded that confined growth of MOF might probably result in partially amorphous regions formed in interfaces which would also have an influence on the permeation performance, similar to the reported ZIF-zeolite cases by Eum and co-workers[64,65]. The membrane displays a slightly higher $H_2/CO_2$ selectivity than $H_2/CH_4$, despite the smaller kinetic diameter of $CO_2$ (0.33 nm) compared with $CH_4$ (0.38 nm). It can be explained from the nitrogen-containing TpPa-1 and ZIF-67 structures which cause a preferential adsorption of $CO_2$ compared to other gases (i.e., $H_2$, $CH_4$) at room temperature[66,67]. Due to this adsorptive interaction, the resistance for $CO_2$ diffusion is increased in mixed-gas permeation, thus simultaneously blocking the molecular transport $H_2$ through the membrane. For comparison, we measured the supported $Co^{2+}$@TpPa-1 layer (only impregnation with Co $(NO_3)_2 \cdot 6H_2O$ solution) and the selectivity of $H_2/CO_2$ mixtures is about 13.4, without substantial improvement in performance compared to the ZIF-67-in-TpPa-1 membrane. This experimental finding is a clear proof that the enhancement of the membrane performance is due to the formation of MOFs not a result of pore blocking by metal ions incorporation.

The effect of confined growth time on the separation performance of the resulting membrane was also investigated by using the equimolar $H_2/CO_2$ mixture. As shown in Supplementary Fig. 15, the selectivity of the membrane increases dramatically within 10 h to a value of about 28, and then slows down in the following 40 h. Simultaneously the $H_2$ and $CO_2$ permeances

decrease gradually and then leveled off. The $H_2/CO_2$ selectivity increases with synthesis time since the $CO_2$ permeance drops stronger than the $H_2$ permeance. This finding reveals the membrane can reach the optimized separation performance within 24 h. Despite the decline of the permeances, the $H_2$ permeance after 48 h synthesis time is still on an ultrahigh value of 3374, suggesting an only partly filling by MOF rather than an entire occupancy of the COFs 1D channel. In this case, there is a small transfer resistance for $H_2$ molecules diffusion during gas permeation. The above result also implies the intrinsic virtue of anti-trade-off behavior in the MOF-in-COF membrane achieving remarkably improved separation selectivity without sacrificing too much the permeance by incorporation of only a small amount (~13.3 vol% calculated based on the Co/O atomic (molar) ratio from XPS, Supplementary Fig. 7) of MOFs in the COF layer.

Considering practical applications, a continuous gas-permeation measurement for equimolar $H_2/CO_2$ and $H_2/CH_4$ mixtures was carried out for over 60 h. As shown in Fig. 3c, the separation performance was scarcely degraded, indicating a good running stability. Moreover, gas permeation was also measured at a higher temperature to evaluate the thermal stability of the membrane. It can be seen that despite the $H_2/CO_2$ separation selectivity gradually decreases from 34.8 to 18 with increasing temperature due to the much more activated diffusion of $CO_2$ compared to $H_2$, the MOF-in-COF membrane (Supplementary Fig. 16) still remains stable at 180 °C for over 100 h. Figure 3e, f illustrate the selectivity of $H_2/CO_2$, and $H_2/CH_4$ as a function of $H_2$ permeability for our MOF-in-COF membranes and other types of membranes reported in the literature (please see the detailed comparison in Supplementary Tables 2, 3). Notably, the MOF-in-COF membranes such as ZIF-67-in-TpPa-1 exhibits high values in both $H_2$ permeance and selectivity in comparison with other membranes. The overall performance surpasses the latest Robeson upper bound limits for polymer membranes[68]. In particular, the ZIF-67-in-TpPa-1 membrane shows state-of-the-art selectivity for $H_2/CO_2$ mixture among the reported COF membranes. The excellent performance together with robust operation stability recommends the MOF-in-COF membranes for advanced $H_2$ purification and production as well as carbon capture.

**Universality demonstration of MOF-in-COF design concept.** For the further demonstration of broad applicability of the MOF-in-COF design concept, we fabricated another two types of MOF-in-COF membranes with a similar protocol as with the ZIF-67-in-TpPa-1 membrane. The ZIF-8-in-TpPa-1 membrane (Supplementary Figs. 17, 18) was prepared via two-stage immersion of the TpPa-1 membrane into the ZIF-8 precursor solution. For the ZIF-67-in-TpBD membrane (Supplementary Figs. 19, 20), a supported TpBD layer (Supplementary Fig. 21) was first synthesized via the reaction of Tp and benzidine (BD) (Supplementary Figs. 22–24)[55], and then followed the two-stage immersion process for the ZIF-67 preparation. As shown in Fig. 4a–h, both ZIF-8-in-TpPa-1 membrane and ZIF-67-in-TpBD membrane are continuously grown on the porous $\alpha$-$Al_2O_3$ substrate and their layer thickness is about 1 µm and 0.9 µm, respectively, from the EDXS mappings (Fig. 4d, h). The bumps appeared on the ZIF-67-in-TpBD membrane surface are consisted of stacked 2D COFs (TpBD), which are formed from the rearrangement of spherical polycrystallites deposited on the surface during the growth of TpBD layer (Supplementary Fig. 25). Gas-separation permeance were also evaluated and the results are shown in Fig. 4i, j. It can be seen that the ZIF-8-in-TpPa-1 membrane has separation factors of 23.1, 22.1, 63.0, and 86.6, for $H_2/CO_2$, $H_2/CH_4$, $H_2/C_3H_6$, and $C_3H_8$ gas mixture, respectively,

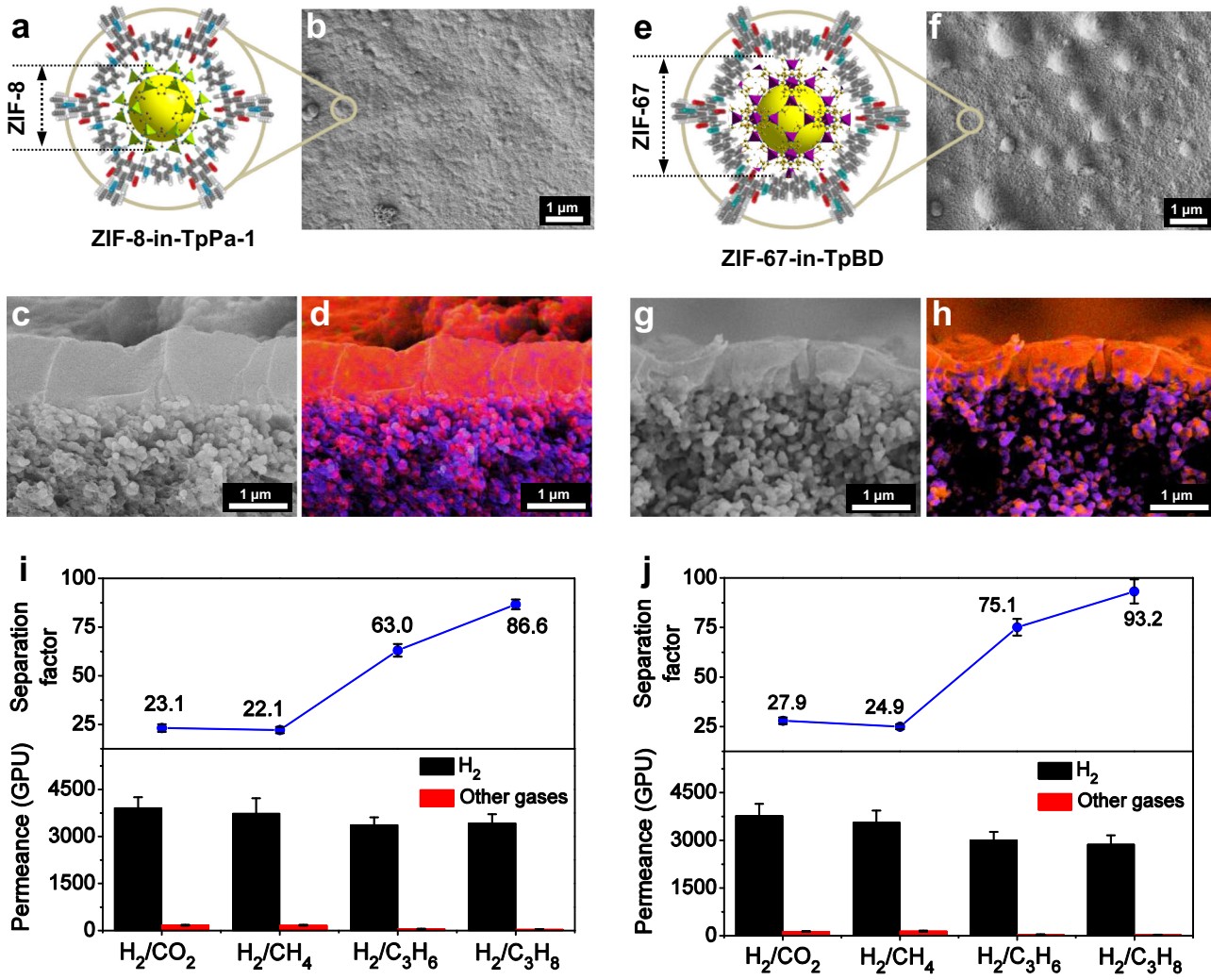

**Fig. 4 MOF-in-COF membrane morphology and gas-permeation performance.** Schematic showing the pore structure (**a**, **e**), top-view (**b**, **f**) and cross-sectional (**c**, **g**) SEM images of ZIF-8-in-TpPa-1 membrane (**a–c**) and ZIF-67-in-TpBD membrane (**e–g**). **d**, **h** EDXS mappings and elemental distributions corresponding to (**c**, **g**). C Kα1_2 signal, red; Al Kα1 signal, purple; Zn Kα1 and Co Kα1 signals, green. $H_2$ permeances and separation factors of (**i**) ZIF-8-in-TpPa-1 membrane and (**j**) ZIF-67-in-TpBD membrane for equimolar binary gases. All measurements at 298 K and 1 bar.

and simultaneously maintains a high $H_2$ permeance of above 3000 GPU. The separation selectivity is not as high as that of the ZIF-67-in-TpPa-1 membrane, probably due to that ZIF-67 has a slightly smaller effective pore aperture than ZIF-8, in consideration of the more rigid Co–N bonds than the Zn–N bonds causing a reduced lattice flexibility due to less ligand flipping motion of the methylimidazolate linker in ZIF-67[56]. Despite this, the overall performance of the ZIF-8-in-TpPa-1 membrane still exceeds the Robeson upper bounds for $H_2/CO_2$ and $H_2/CH_4$ (Fig. 3e, f). In addition, the ZIF-67-in-TpBD membrane also displays superior sieving performance for selective $H_2$ separation. For example, the separation factor for $H_2/CO_2$ gas pair can reach 27.9 much higher than that of the supported TpBD layer without ZIF-67 (Supplementary Fig. 26). These results further suggest the formation of MOF-in-COF pore structures inside the membranes.

## Discussion

The formation of a MOF-in-COF structure depends on multiple factors including capillary action, coordination of metal ions to COF matrix/organic ligands, and internal diffusion of organic ligands, as illustrated in Supplementary Fig. 27. First, driven by the capillary action, $Co^{2+}$ ions companied with the solvent methanol enter the 1D channels of COF, and can coordinate with the N

atoms in the pore wall of the COF tube. Afterward, the added 2-meIM diffuse through the COF pores due to the concentration gradients, encounter and coordinate with the $Co^{2+}$ having been located already inside the 1D channel. Through a rapid nucleation and complete confined growth, ZIF-67 has been formed inside the COF pores, and thereby the MOF-in-COF structure is formed. It's worth noting that the pre-formed MOF near the orifice could hinder the follow-up 2-meIM diffusion into the interior. Therefore, the MOF is most probably grown close to the orifice of 1D channels of the COF, not homogeneously in the entire COF layer. It is indeed difficult to detect the entry depth and specific shape of the MOF inside the COF pore, because of its tiny dimension in the range of one or a few unit cells and since the MOF species are highly dispersed in the COF matrix. Despite this, it plays an important role in the transport of gas molecules. In addition, it is also found that straight one-stage immersion into the as-prepared MOF precursor solution did not induce a significant improvement in membrane performance. This observation is mainly due to the fact that the fast-forming MOF nanocrystals in solution cannot get into the COF pore caused by the steric hindrance effect.

To elucidate the gas-separation mechanism, molecular dynamics (MD) simulation was performed to investigate the gas-permeation behavior of equimolar $H_2/CO_2$ and $H_2/CH_4$ mixtures through the

COF membrane and MOF-in-COF membrane (Supplementary Figs. 28, 29), respectively. It is observed that 60% of $H_2$ molecules permeated through the TpPa-1 membrane within 100 ps of the simulation duration, but also accompanied by the permeation of 27% of $CO_2$ molecules (Supplementary Fig. 28a, b). In contrast, 57% of $H_2$ molecules could permeate through the ZIF-67-in-TpPa-1 membrane, while only 3% $CO_2$ molecules could penetrate at the same time (Supplementary Fig. 28c, d). Moreover, most $CO_2$ molecules were adhered to the ZIF-67-in-TpPa-1 membrane surface, suggesting the confined transport channels for $H_2$ molecules existed in the MOF-in-COF micro/nanopore network, which resulted in remarkable selectivity improvement excellent separation selectivity. A similar phenomenon was also observed during the simulation for the equimolar $H_2/CH_4$ mixture. As shown in Supplementary Fig. 29, 47% of $CH_4$ molecules could permeate through the COF membrane, which is far higher than that of only 20% $CH_4$ of molecules through the ZIF-67-in-TpPa-1 membrane within the same time. The simulation results also indicate that besides the molecular sieving effects of the MOF-in-COF micro/nanopores, the low $CO_2$ permeance is also attributed by the retarded diffusivity because of the nitrogen-containing TpPa-1 and ZIF-67 structure with adsorption for $CO_2$. This is consistent with the experimental finding that the separation selectivity for $H_2/CO_2$ mixtures is relatively higher than that for the $H_2/CH_4$ mixtures.

In conclusion, we have explored a MOF-in-COF assembly strategy to design advanced molecular sieving membranes. The formation of a unique MOF-in-COF micro/nanopore network in a selective layer endows smaller effective pore size and more precise molecular sieving properties, which leads to the enhanced separation selectivity of $H_2$ from other more bulky gases. Meanwhile, the vertical 1D channel in 2D COFs unlike the zigzag-type pores with smaller resistances enables the ultrafast transfer of $H_2$ molecules through the membranes. Owing to the synergy between different nanoporous materials in the MOF-in-COF layer, the resulting membranes exhibit ultrahigh $H_2$ permeance and remarkable enhancement in separation selectivity for gas mixtures as compared to individual COF and MOF membranes. Moreover, the overall performances far exceed the latest Robeson upper bounds for $H_2/CO_2$ and $H_2/CH_4$, and are superior to other gas-separation membranes. The excellent performance combined with a high stability recommend the MOF-in-COF membranes as promising candidates for practical $H_2$ purification and production, $CH_4$ reforming process as well as $CO_2$ capture and utilization. Considering the versatility of successful fabrication of various MOF-in-COF membranes, our study not only provides an intriguing pore-engineering concept for COFs, but also recommends MOF/COF-based composites for energy and environment-relevant separation processes.

## Methods

**Preparation of supported TpPa-1 layer and TpBD layer**. The supported TpPa-1 layer was prepared on a porous asymmetric $\alpha$-$Al_2O_3$ disk substrate (18 mm in diameter, 1 mm in thickness, 70 nm pore size in the top layer, from Fraunhofer IKTS, Germany) via a facile protocol of in situ solvothermal synthesis method. First, the porous $\alpha$-$Al_2O_3$ disk surface was activated by HCl solution (1 M), and then amino-modified with 3-aminopropyltriethoxysilane (APTES) (2 mM in toluene) at 110 °C for 2 h under nitrogen atmosphere. The amino-$Al_2O_3$ disk surface was grafted with aldehyde groups by using the 1,3,5-triformylbenzene (TFB) dioxane solution (3 mg mL$^{-1}$) at 150 °C for 1 h. After washing with ethanol, the aldehyde-$Al_2O_3$ disk was fixed onto a PTFE holder and placed face down into a 25.0 mL Teflon-lined stainless steel vessel. Then the TpPa-1 precursor solution (31.5 mg 1,3,5-triformylbenzene (Tp) and 24 mg $p$-phenylenediamine (Pa-1) in 6 mL mixed solvent of dioxane/mesitylene (1:1, v/v) with the presence of 1 mL 3 M acetic acid solution) was poured and followed by a nitrogen injecting for 5 min. The Teflon-lined stainless steel vessel was sealed and placed into an oven at 120 °C for 72 h. The supported TpPa-1 layer was obtained after a thorough washing with dioxane, ethanol, and a drying treatment. With a similar procedure, the supported TpBD layer could be synthesized onto the aldehyde-$Al_2O_3$ substrate by using the

precursor solution (31.5 mg Tp and 41.5 mg benzidine (BD) in 6 mL mixed solvent of dioxane/mesitylene (1:1, v/v) with the presence of 1 mL 3 M acetic acid solution).

**Preparation of MOF-in-COF membranes**. MOF-in-COF membranes including ZIF-67-in-TpPa-1 membrane, ZIF-8-in-TpPa-1 membrane, and ZIF-67-in-TpBD membrane were prepared by a two-stage immersion process. Taking the synthesis of ZIF-67-in-TpPa-1 membrane as an example, the supported TpPa-1 layer was first fixed face out onto a PTFE holder and then vertically dipped into a Co(NO$_3$)$_2$·6H$_2$O solution (27.6 mg in 5 mL methanol) at room temperature for 24 h. The vertical placement is to avoid the deposition of ZIF-67 nanocrystals on the membrane surface. Afterward, a 2-meIM solution (30.75 mg in 5 mL methanol) was added, and the formed blue solution was allowed to stand for another 24 h. The ZIF-67-in-TpPa-1 membrane was thoroughly washed with methanol to remove the residual Co$^{2+}$ and 2-meIM as well as the possible ZIF-67 nanocrystals adhered to the surface and then dried at 100 °C overnight. The ZIF-8-in-TpPa-1 membrane was obtained by immersing the supported TpPa-1 layer in a zinc nitrate hexahydrate solution (36.65 mg in 5 mL methanol) for 24 h, and then was continuously immersed for another 24 h after the addition of the 2-meIM solution (81.1 mg in 5 mL methanol). The ZIF-67-in-TpBD membrane was obtained through two-stage immersion of the supported TpBD layer into a Co(NO$_3$)$_2$·6H$_2$O solution and the ZIF-67 precursor solution with the same concentration as the synthesis of ZIF-67-in-TpPa-1 membrane.

**Preparation of pure MOF membranes**. Pure ZIF-67 membrane was prepared via a seeded-assisted in situ growth approach. First, ZIF-67 nanoseeds were prepared at room temperature. In a typical synthesis, 0.546 g of Co(NO$_3$)$_2$·6H$_2$O and 0.616 g of 2-meIM were dissolved in a 15 mL of methanol, respectively. After blending of the two methanolic solutions and sonication for 15 min, the dark-purple precipitate was collected by centrifugation, and washed by methanol for several cycles. The obtained ZIF-67 nanoseeds colloid without drying was directly dispersed in a 0.1 g PEI (50 wt% in water) aqueous solution (4 mL) in presence of 10 mg NaHCO$_3$, and then treated under ultrasonic conditions for 10 min. Afterward, an activated $\alpha$-$Al_2O_3$ disk substrate was dip-coated in the above seeding solution for 20 s, and dried in air. If needed, this procedure could be repeated to ensure a satisfied coverage of ZIF-67 seeds. For the synthesis of the ZIF-67 membrane, the seeded $\alpha$-$Al_2O_3$ disk was placed vertically in a 50 mL autoclave, which was filled with synthesis solution (0.11 g of Co(NO$_3$)$_2$·6H$_2$O and 2.27 g of 2-meIM in a mixed solvent of 2.5 ml of methanol and 17.5 ml of D.I. water). The autoclave was kept in an oven at 120 ºC for 48 h. After natural cooling, the as-prepared membrane was washed with methanol and dried at 80 ºC overnight.

**Characterization**. Observation on the morphologies of samples was carried out by using a JEOL JSM-6700F instrument with a cold field emission gun operating at 2 kV and 10 mA. All samples were coated with a 15-nm thick gold layer by a vacuum sputtering before measurement to enhance the conductivity. Energy-dispersive X-ray spectroscopy (EDXS) mapping and elemental analysis were performed on the scanning electron microscopy (SEM) at 15 kV, 10 mA and 15 mm lense distance. The TEM measurements were accomplished with JEOL JEM-2100F transmission electron microscope operated at an accelerating voltage of 200 kV. XPS spectra were recorded on a Thermo Scientific K-Alpha+ spectrometer using Al K$\alpha$ radiation (12 kV, 6 mA) as the energy source. The pressure in the instrumental chamber was less than $2 \times 10^{-7}$ mPa. No radiation damage was observed during the data collection time. Binding energies were calibrated on C1s (284.8 eV). The X-ray diffractometer (XRD) patterns were recorded on a Bruker D8 Advance diffractometer (Cu K$_\alpha$ X-ray radiation, $\lambda = 1.54$ Å) at room temperature. Each XRD pattern was acquired ranging from 3° to 35° at a rate of 0.01° s$^{-1}$ at a voltage of 40 kV and current of 40 mA. The attenuated total reflectance-Fourier transform infrared spectra (ATR-FTIR, 400–4000 cm$^{-1}$; resolution of 0.4 cm$^{-1}$) were obtained by using a spectrometer (Agilent Technologies Cary 630 FTIR). To detect the FTIR spectrum of the membrane samples, the selective separation layer was shaved off as powder for measurement. N$_2$ adsorption–desorption measurements were performed at 77 K by using a Micromeritics ASAP2460 surface area and pore distribution analyzer instrument. Powdered samples were vacuum degassed at 120 °C for 10 h before the adsorption experiments. The resulting isotherms were analyzed using the Brunauer–Emmett–Teller (BET) method and the $t$-plot micropore volume method.

**Gas-separation measurement**. The prepared membrane was fixed in a home-made gas-permeation apparatus (Fig. 3a) sealed with rubber O-rings. For the single-gas-permeation measurement, both feed and sweep flow rates were set to 50 mL min$^{-1}$. The N$_2$ was used as the sweep gas and the pressures at both sides were kept at 1 bar. For the mixed-gas-permeation measurement, a series of equimolar (1:1) binary gas mixture such as H$_2$/CO$_2$, H$_2$/CH$_4$, H$_2$/C$_3$H$_6$, and H$_2$/C$_3$H$_8$ were applied to the feed side of the membrane, and the feed flow rate was kept constant at 50 mL min$^{-1}$ (each gas of 25 mL min$^{-1}$). The pressures at both sides were still constant at 1 bar during the measurement process and the N$_2$ (50 mL min$^{-1}$) was used as the sweep gas. A calibrated gas chromatograph (HP 6890B) was used to detect the component concentration on the permeate side after the measurement system ran stable.

The permeance of component $i$ ($P_i$) was calculated as follows (Eq. 1):

$$P_i = \frac{N_i}{A\Delta p_i} = \frac{F_i}{\Delta p_i}, \tag{1}$$

where $N_i$ is the permeation rate of component $i$ (mol s$^{-1}$), $\Delta p_i$ is the partial pressure difference of component $i$ (Pa), and $A$ is the membrane area (m$^2$). $F_i$ denotes the molar flux of component $i$ (mol m$^{-2}$ s$^{-1}$). Every permeance was calculated by the average of five data points. The unit GPU is used for the gas permeance (1GPU = 3.3928 × 10$^{-10}$ mol m$^{-2}$ s$^{-1}$ Pa$^{-1}$).

The selectivity of two components in the single-gas permeation (or ideal selectivity) was calculated as follows:

$$\alpha = \frac{P_i}{P_j}. \tag{2}$$

The selectivity ($\alpha_{i,j}$) of an equimolar binary gas mixture (or separation factor) is calculated as follows (Eq. 3):

$$\alpha_{i,j} = \frac{y_i/y_j}{x_i/x_j}, \tag{3}$$

where $x$ and $y$ are the molar fractions of the corresponding component $i, j$ in the feed and permeate side, respectively.

**Maxwell prediction of gas permeation**. Considering the near-spherical shape of the MOF dispersed in the COF layer, we using the Maxwell model[63] to simply estimate and quantitatively analyze the separation performance of MOF-in-COF membrane for equimolar H$_2$/CH$_4$ mixture. The Maxwell model (Eq. 4) describes the mixed matrix membrane permeance in terms of the permeance of the individual phases and the volume fraction of the dispersed phase:

$$P_{\text{MOF-in-COF}} = P_{\text{COF}} \frac{2(1-\varphi) + \lambda(1+2\varphi)}{(2+\varphi) + \lambda(1-\varphi)}, \tag{4}$$

where $P_{\text{MOF-in-COF}}$ is the MOF-in-COF membrane gas permeance, $\varphi$ is the volume fraction of the MOF, and $\lambda$ is the permeance ratio of the two phases, $P_{\text{MOF}}/P_{\text{COF}}$, which could be obtained by the adsorption and gas-permeation data.

**MD simulations**. Supplementary Fig. 28a shows the eclipsed atomic structure of TpPa-1 ($a = b = 22.556$ Å) composing of five-layered nanosheets with a thickness of approximately 1.5 nm. The atomic structure of ZIF-67-in-TpPa-1 is built by incorporating one ZIF-67 unit cell into one TpPa-1 pores, as illustrated in Supplementary Fig. 28c. Molecular dynamics simulation was carried out by Materials Studio software 6.0 package with COMPASS force field[69,70]. The dimension of the simulation box was set to 45.1 Å × 45.1 Å × 166.5 Å. There were two chambers containing an equimolar mixture of H$_2$/CO$_2$ or H$_2$/CH$_4$ (30 molecules for each component) on the left and a vacuum on the right, respectively, which were separated by the TpPa-1 membrane or ZIF-67-in-TpPa-1 membrane approximately in the middle along the $z$-axis. The system was optimized before diffusion simulation. NVT (constant particle number, volume, and temperature) ensemble was employed for simulation. The initial velocities were random, and the Andersen thermostat was employed to maintain a constant simulation temperature of 298.0 K. The MD simulation was performed for 20–100 ps with a time step of 1 fs using the Forcite module.

## Data availability
All data that support the findings of this study are available within the paper and its Supplementary Information or from the corresponding author upon reasonable request. Source data are provided with this paper.

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

## Acknowledgements

This work was financially supported by the Deutsche Forschungsgemeinschaft (DFG, Ca 147/21) and National Natural Science Foundation of China (Program No. 51773012). H.F. is grateful for the financial support from Alexander von Humboldt Foundation.

## Author contributions

H.F. and J.C. conceived the research. H.F. designed and performed experiments. A.M. and I.S. were in charge of gas-chromatography and analytics. All authors contributed to analysis and discussion on the data. M.P and H.M constructed the molecular models and conducted the MD simulations. H.F. and J.C. wrote and edited the manuscript. All authors reviewed and commented on the paper.

## Funding

## Competing interests

The authors declare no competing interests.
