## [Peer Review File · Nature Communications]

REVIEWER COMMENTS

Reviewer #1 (Remarks to the Author):

In the manuscript "MOF-in-COF molecular sieving membrane for selective hydrogen separation", the authors introduced a simple method for synthesizing MOFs in a continuous two-dimensional COF membrane layer to prepare a MOF-in-COF membrane and its application in selective gas separation. The prepared membrane has a unique MOF-in-COF micro/nanopore network and exhibits excellent performance in terms of ultra-high hydrogen permeability and separation selectivity for mixed gas. Nevertheless, I do have some suggestions and comments on this manuscript that I think would help to increase the impact of this manuscript. The detailed comments are as follows:

1. In this work, does the growth of MOF in a two-dimensional COF layer depend on some interaction force?
2. It is best to briefly state in the conclusion why MOF-in-COF has an extremely high hydrogen permeability and significantly enhanced ability to separate gas mixtures compared to MOF and COF.
3. How are the shallow pits on the surface shown in the SEM image of MOF-in-COF in Figure 4f generated?
4. The format of the references is not completely consistent, such as Ref [3]、 Ref [23] and Ref [25], please check it carefully.
5. Some articles should be cited. For example, EnergyChem 2019, 1, 100005; EnergyChem 2019, 1, 100006; EnergyChem 2019, 1, 100016.

Reviewer #2 (Remarks to the Author):

This manuscript presents the fabrication of metal organic framework (MOFs) inside a supported COF layer to prepare MOF-in-COF membranes. The resulting MOF-in-COF membranes exhibit high separation performance for H₂/CO₂ and H₂/CH₄ surpasses the Robeson upper bounds. The novelty of this work mainly comes from the new-type MOF-in-COF membranes they prepared for gas separation. Unfortunately, I cannot find any direct proofs of ZIFs synthesized inside the confined channel of COF layer.

1. The authors claimed ZIF-67 were synthesized in the channel of TpPa-1. The PXRD and IR show no peaks can be assigned to ZIF-67. The XPS show some circumstantial evidence from the binding energy from Co2p and Co-N. This is the evidence that the adsorbed Co and 2-meIM did form some coordination compounds. The question is how to prove the coordination compounds are ZIF-67? Can

the authors prove that there is no any chance to synthesize any other compounds except for ZIF-67 in the nano-sized COF channel?

2. In supplementary Figure 9, the yellow zoom drawn by the authors will mislead the readers. In fact, the "edge" of the "ZIF" cannot be seen and this is not sufficient to prove the formation of ZIF-67

3. Figure 2f and supplementary Figure 21 are misleading. If all channel of COFs were occupied by "ZIF", why the content of Co is only about 2 atomic %? How many channels were occupied and does the content of "ZIF" affect the performance of gas separation?

In conclusion, I cannot recommend its publication in Nature Communications.

Reviewer #3 (Remarks to the Author):

Interesting concept and fabrication process. The separation performance is impressive and indeed relevant for challenging gas separations. The characterization is quite comprehensive and elucidates most of the structural features of interest. I would recommend publication in NCOMMS with the following suggestions for revision:

- There could be more discussion on the influence of the MOF-COF interfaces. This is clearly important considering the 1D nature of the COF pores: an unfavorable connection between the MOF and COF could completely shut down permeation. I understand it is extremely difficult to characterize it in detail, but some additional comments are desirable. The authors discuss partial pore blocking, can this be connected to the interfacial structure?

- What is the estimated "loading" (in vol or wt%) of the COF sheets in the MOF matrix in the final membranes ? Any comment on how this can be varied?

- Also, it is very simple to classically estimate the performance of a membrane containing sheets/flakes (Cussler-type models). In the limit of high aspect ratio flakes, it reduces essentially to resistance in series model. Disagreement of the experimental separation results with Cussler-type model is usually a good sign that interface effects are in play.

- The authors compare the COF-MOF membrane to a pure COF membrane, but they could also analyze the COF-MOF membrane in relation to a pure MOF membrane? Is the performance of a pure ZIF-67 membrane for H₂/CO₂ and H₂/hydrocarbon separations known (or predicted from simulations)? A brief discussion on the advantage of spacing the ZIF-67 regions with COF sheets could be given.

- It may be worth mentioning recent publications on MOF-zeolite nanosheet membranes (e.g., Eum et al, ACS Appl Mater Interf, 2020) as a similar concept. The above paper also states questions regarding the interfaces and MOF crystallization. In the present context, is the ZIF-67 crystallized between the COF sheets the same as bulk ZIF-67 (e.g., rigidity of the framework) ? These questions do not need to be answered in this paper, but stating them in connection to the ZIF-zeolite case would be useful for the community.

Reviewer #1 (Remarks to the Author):

General comments:

In the manuscript "MOF-in-COF molecular sieving membrane for selective hydrogen separation", the authors introduced a simple method for synthesizing MOFs in a continuous two-dimensional COF membrane layer to prepare a MOF-in-COF membrane and its application in selective gas separation. The prepared membrane has a unique MOF-in-COF micro/nanopore network and exhibits excellent performance in terms of ultra-high hydrogen permeability and separation selectivity for mixed gas. Nevertheless, I do have some suggestions and comments on this manuscript that I think would help to increase the impact of this manuscript. The detailed comments are as follows:

Response to general comments: Many thanks for the reviewer's comments and high affirmation of our research work. According to the reviewer's suggestion, we explained in detail the concerns mentioned by the reviewer, such as the MOF growth in a COF layer, and the reason for the formation of "shallow pits" (actually they are the bumps not shallow pits) in the surface SEM image in **Figure 4f**. The reasons accounting for the more superior performance of MOF-in-COF membrane than the individual MOF and COF membranes have been briefly stated and added into the conclusion chapter. Moreover, we have carefully checked out the format of all references and corrected them in the revised manuscript. Also some more relevant articles mentioned by the reviewer have been cited in the reference. Please see the detailed point-by-point responses as below.

Comment 1: In this work, does the growth of MOF in a two-dimensional COF layer depend on some interaction force?

Response 1: Yes, in this work, formation of MOF inside a two-dimensional COF layer depends on multiple factors including capillary action, coordination of metal ions to COF matrix/organic ligands, and internal diffusion of organic ligands. Take the ZIF-67 in the TpPa-1 layer, for instance, as illustrated in **Fig. R1 (Supplementary Fig. 27)**. First, driven by the capillary action, Co^{2+} ions accompanied with the solvent of methanol enter the 1D channels of COF, and can coordinate with the N elements in COF matrix. Afterwards, the added 2-meIM diffuse towards the inside COF pores due to the concentration gradients, encounter and coordinate with the Co^{2+} having been located the inside. Through a rapid nucleation and complete confined growth, the ZIF-67 is synthesized inside the COF pores, and thereby the MOF-in-COF structure is formed. It's worth noting that the pre-formed MOF near the orifice could hinder the diffusion of 2-meIM into the depths. Therefore, the MOF is most probably grown close to the orifice of 1D channels of COF, not in the entire COF layer. More detailed description of the MOF growth in the 2D COF layer has been added into the revised manuscript.

Fig. R1 Schematic illustrating the formation process of MOF-in-COF structure.

Comment 2: It is best to briefly state in the conclusion why MOF-in-COF has an extremely high hydrogen permeability and significantly enhanced ability to separate gas mixtures compared to MOF and COF.

Response 2: According to the reviewer’s suggestion, we have further stated in the conclusion chapter relevant reasons accounting for the MOF-in-COF membrane with superior performance to that of the individual MOF and COF membranes in the revised manuscript. Please see the revised part as follows:

The formation of unique MOF-in-COF micro/nanopore network in selective layer endows smaller effective pore size (probably as a result of the greater rigidity of the confined MOF lattice) and more precise molecular sieving properties, which leads to the enhanced separation selectivity of H₂ from other more bulky gases. Meanwhile, the vertical 1D channel in 2D COFs unlike the zigzag-type pores with smaller resistances enables the ultrafast transfer of H₂ molecules through the membranes. Owing to the synergy between different nanoporous materials in MOF-in-COF layer, the resulting membranes exhibit ultrahigh H₂ permeance and remarkable enhancement in separation selectivity for gas mixtures as compared to individual COF and MOF membranes.

Comment 3: How are the shallow pits on the surface shown in the SEM image of MOF-in-COF in Figure 4f generated?

Response 3: That’s an interesting question. More precisely, on the surface in SEM image of **Figure 4f**, they are not “shallow pits” but bumps consisting of stacking 2D COFs (TpBD) nanosheets (**Fig. R2a and b as below**).

Actually, these bumps are generated during the synthesis of TpBD layer, which is illustrated in the schematic of **Fig. R2c**. Initially, the reactant molecules of 1,3,5-triformylbenzene (Tp) and benzidine (BD) in solution are adsorbed to the aldehyde-modified α -Al₂O₃ surface. Due to the lower interfacial energy and thermodynamic barrier, it would lead to kinetically faster surface nucleation and promote in plane lateral growth of TpBD^[R1]. In this process, in the solution the un-adsorbed monomers are also nucleated and grown into the multilayers either through stacking of metastable nanosheets, or through template growth. Once crystallites are formed, these 2D multilayers are easily aggregated into the polycrystallites in shape of near-spherical structure under the effect of surface tension and interaction between multilayers^[S2-S4]. Subsequently, these solution-grown polycrystallites consisting of 2D TpBD nanosheets adhere or deposit on top of previous COF layers. Furthermore, through the error-checking and self-correction from the templating effect of the underlying surface layers, they are allowed to rearrange and grow as new layers along the surface normal. But the spherical polycrystallites which have not

enough time to completely reinstate the in-plane orientation to the substrate will form the bumps on the surface. We have added some explanations to the main text of the revised manuscript, and also the schematic of **Fig. R2c** as **Supplementary Fig. 25** with the above descriptions was added to the revised Supplementary Information.

Added references to the Supplementary Information (numbered as Ref [S1], Ref [S2], Ref [S3], Ref [S4]).

[R1] Wang, H., He, B., Liu, F., Stevens, C., Brady, M. A., Cai, S., Wang, C., Russell, T. P., Tan, T. W. & Liu, Y. Orientation transitions during the growth of imine covalent organic framework thin films. *J. Mater. Chem. C* **5**, 5090 (2017).

[R2] Wang, S., Zhang, Z., Zhang, H., Rajan, A. G., Xu, N., Yang, Y., Zeng, Y., Liu, P., Zhang, X., Mao, Q., He, Y., Zhao, J., Li, B. G., Strano, M. S. & Wang, W. J. Reversible Polycondensation-Termination Growth of Covalent-Organic-Framework Spheres, Fibers, and Films. *Matter* **1**, 1592-1605 (2019).

[R3] Koo, B. T., Heden, R. F. & Clancy, P. Nucleation and growth of 2D covalent organic frameworks: polymerization and crystallization of COF monomers. *Phys. Chem. Chem. Phys.* **19**, 9745-9754 (2017).

[R4] Smith, B. J. & Dichtel, W. R. Mechanistic Studies of Two-Dimensional Covalent Organic Frameworks Rapidly Polymerized from Initially Homogenous Conditions. *J. Am. Chem. Soc.* **136**, 8783-8789 (2014).

Fig. R2 SEM images of (a) ZIF-67-in-TpBD membrane surface and (b) as-synthesized TpBD powders in the same autoclave as the TpBD membrane. (c) Schematic illustration of formation process of TpBD bumps on the surface.

Comment 4: The format of the references is not completely consistent, such as Ref [3] and Ref [23] and Ref [25], please check it carefully.

Response 4: The Ref [3]; Ref [23] and Ref [25] have been corrected and renumbered in the revised manuscript (as below). Moreover, we also carefully checked the format of all other references and corrected them.

Corrected references in the revised manuscript (Ref [23] and Ref [25] were renumbered as [25], [27], respectively).

[3] Park, H. B., Kamcev, J., Robeson, L. M., Elimelech, M. & Freeman, B. D. Maximizing the right stuff: The

trade-off between membrane permeability and selectivity. *Science* **356**, 1137-1147 (2017).

[25] Diercks, C. S. & Yaghi, O. M. The atom, the molecule, and the covalent organic framework. *Science* **355**, 923-930 (2017).

[27] Slater, A. G. & Cooper, A. I. Function-led design of new porous materials. *Science* **348**, 988-1000 (2015).

Comment 5: Some articles should be cited. For example, EnergyChem 2019, 1, 100005; EnergyChem 2019, 1, 100006; EnergyChem 2019, 1, 100016.

Response 5: We have cited the relevant articles and numbered as Ref [12], Ref [13], and Ref [67] in references.

Added references in the revised manuscript.

[12] Li, D., Xu, H. Q., Jiao, L. & Jiang, H. L. Metal-Organic Frameworks for Catalysis: State-of-the-Art, Challenges, and Opportunities. *EnergyChem* **1**, 100005 (2019).

[13] Li, H., Li, L., Lin, R. B., Zhou, W., Xiang, S., Chen, B. & Zhang, Z. Porous Metal-Organic Frameworks for Gas Storage and Separation: Status and Challenges. *EnergyChem* **1**, 100006 (2019).

[67] Zhou, D. D., Zhang, X. W., Mo, Z. W., Xu, Y. Z., Tian, X. Y., Li, Y., Chen, X. M. & Zhang, J. P. Adsorptive separation of carbon dioxide: from conventional porous materials to metal-organic frameworks. *EnergyChem* **1**, 100016 (2019).

Reviewer #2 (Remarks to the Author):

General comments:

This manuscript presents the fabrication of metal organic framework (MOFs) inside a supported COF layer to prepare MOF-in-COF membranes. The resulting MOF-in-COF membranes exhibit high separation performance for H₂/CO₂ and H₂/CH₄ surpasses the Robeson upper bounds. The novelty of this work mainly comes from the new-type MOF-in-COF membranes they prepared for gas separation. Unfortunately, I cannot find any direct proofs of ZIFs synthesized inside the confined channel of COF layer.

Response to general comments: Many thanks for the reviewer's comments and suggestions. To further confirm the ZIF-67 formed inside the confined channel of COF layer, we provided a new XRD measurement of the MOF-in-COF powders scraped from the alumina substrate by optimizing the test parameters. Moreover, more discussion on this result has been added to the revised manuscript. According to the reviewer's suggestion, the yellow marks in **supplementary Figure 9 (revised supplementary Fig. 10)** have been deleted, and also **Figure 2f** as well as **supplementary Figure 21 (revised supplementary Fig. 27)** has been repainted in the revised manuscript. In addition, we prepared more MOF-in-COF membranes with different synthesis time and conducted the gas permeation measurement, aiming at investigating the effect of synthesis time (or MOF content) on the separation performance. Please see the detailed point-by-point responses as follows.

Comment 1: The authors claimed ZIF-67 were synthesized in the channel of TpPa-1. The PXRD and IR show no peaks can be assigned to ZIF-67. The XPS show some circumstantial evidence from the binding energy from Co2p and Co-N. This is the evidence that the adsorbed Co and 2-meIM did form some coordination compounds. The question is how to prove the coordination compounds are ZIF-67? Can the authors prove that there is no any chance to synthesize any other compounds except for ZIF-67 in the nano-sized COF channel?

Response 1: Thanks for the reviewer's comments. It is really not easy to detect and prove the coordination compounds are ZIF-67 because of the tiny dimension and low amount of MOFs formed in the channel of TpPa-1. Over the past one more month, we even tried the high-resolution scanning tunneling microscopy (STM) to intuitively observe the morphology of single MOF-in-COF pore, but the result was still not satisfactory (please see the captured STM images of **Fig. R3** as below). This is because the compact MOF-in-COF layer is difficult to be exfoliated as single- or several-atomic layer in thickness suitable for the STM characterization. Thus, the captured STM images do not show the obvious and regular COF pore or MOF-in-COF pore structures, but only the agglomerates consisting of the self-curved COF nano-sheets.

Fig. R3 STM images of the MOF-in-COF layer exfoliated in (a) ethanol and (b) tetrahydrofuran. (The STM was performed at the Institute of Chemistry, Chinese Academy of Sciences (CAS), Beijing).

After discussion with all co-authors, it is generally considered that perhaps the XRD is the most possible approach to directly prove the coordination compound as ZIF-67. Thus, different from the previously conducted XRD characterization of supported MOF-in-COF membrane (**Figure 2a**), we re-measured the XRD patterns of the powdered MOF-in-COF samples scraped from the α -Al₂O₃ substrate. This can avoid the disturbance of strong XRD diffraction from the α -Al₂O₃ substrate. To intensify the XRD diffraction peaks of ZIF-67 as much as possible, the samples with enough amounts was needed and collected from at least 10 MOF-in-COF membranes in parallel. To ensure the accuracy, the scraped MOF-in-COF powders were divided into three groups for the XRD measurement. By optimizing the test parameters, it is expected that some XRD reflections related to ZIF-67 could be captured. The measurement result is shown in **Fig. R4**. It can be seen that there are several diffraction peaks assigned to the ZIF-67 appearing in the magnified XRD zone, despite they are still not very strong. This clearly indicates the formation of ZIF-67 inside the COF layer. It should be noted that the relative diffraction intensity of the peaks in three groups seems a little different. This phenomenon is explainable because the crystalline orientation in MOF-in-COF membrane could be destroyed after scraping and the subsequent grinding treatment.

Fig. R4 XRD patterns of MOF-in-COF powders scraped from the substrate with the insets of scraping photograph (left) and magnified zone (right). To ensure the measurement accuracy, the MOF-in-COF powers were collected from at least 10 parallel membrane samples, and then divided into three (1, 2, 3) parts for use.

In addition, theoretically, the formed coordination compounds are most likely the ZIF-67, because this type of

MOF are easily synthesized via the reaction between $\text{Co}(\text{NO}_3)_2 \cdot 6\text{H}_2\text{O}$ and 2-meIM by using the regular molar proportions at room temperature. Of course, considering the weak XRD diffractions, we can not exclude the coordination compounds contain few amorphous ZIF-67 formed in the confined COF layer. Even so, the formed microporous network in the nano-sized COF channel is still conducive to the improvement in the hydrogen separation selectivity compared to the individual COF membrane.

Anyway, the emerged ZIF-67 diffraction peaks combined with other supporting evidence such as XPS and adsorption-desorption isotherms could confirm the ZIF-67 synthesized in the COF layer. We certainly appreciate and respect the reviewer's comment, and this is really an inspiring suggestion. We are attempting to synthesize new MOF-in-COF composites by using larger-pore 2D COFs (with a pore aperture of ~ 5 nm). Hopefully, in our next research work we could elucidate the specific shape of MOF formed in the confined 1D channel of 2D COF and the growth mechanism.

Some of the above discussion has been added into the revised manuscript, and **Fig. R4** as **Supplementary Fig. 4** was added to the Supplementary Information.

Comment 2: In supplementary Figure 9, the yellow zoom drawn by the authors will mislead the readers. In fact, the edge of the ZIF-67 cannot be seen and this is not sufficient to prove the formation of ZIF-67

Response 2: The yellow marks in supplementary **Figure 9** (revised supplementary **Fig. 10**) have been deleted, and the relevant statement has been revised in the revised manuscript.

Comment 3: Figure 2f and supplementary Figure 21 are misleading. If all channel of COFs were occupied by ZIF-67, why the content of Co is only about 2 atomic %? How many channels were occupied and does the content of ZIF-67 affect the performance of gas separation?

In conclusion, I cannot recommend its publication in Nature Communications.

Response 3: Many thanks for your comments. We have revised and repainted the **Figure 2f** and **supplementary Figure 21** (revised supplementary **Fig. 27**). Yes, it is certain that not all channels of COFs were occupied by MOFs. During the confined growth of MOFs, the pre-formed MOF near the orifice could hinder the diffusion of 2-meIM into the depths. Therefore, the MOF is most probably grown close to the orifice of 1D channels of COF, not in the entire COF layer.

It is difficult to determine how much space was occupied by the MOFs and the specific distribution position in the COF channels. But theoretically, at least, each COF channel in the selective layer has MOFs, even if only one MOF unit cell. Otherwise, the separation performance of the MOF-in-COF membrane could not be significantly boosted. To confirm this, we further investigated the effect of synthesis time on the separation performance. As shown in **Fig. R5**, the separation selectivity of the H_2/CO_2 mixture increased dramatically within 10 h after addition of 2-meIM, and then slowed down after 24 h. The H_2 permeance decreased gradually and then leveled off. For example, the H_2/CO_2 selectivity was only 12.5 for the pristine $\text{Co}^{2+}@$ TpPa-1 membrane (without addition of

2-meIM). It has a more than two-fold improvement (28.2) for the MOF-in-COF membrane with a synthesis time of only 10 h, but subsequently, the selectivity increased not too much, only from 33 to 35 by extending the synthesis time from 24 h to 48h. This suggests the saturation growth of MOF within 24 h and the MOF-in-COF pore structure have been formed in each 1D channel of COFs. Despite the decline, the H₂ permeance could still remain at an ultrahigh value of 3374 after 48 h, suggesting a low content of MOF incorporation without filling up the entire 1D channel of COF which has a small transfer resistance for H₂ molecules diffusion during gas permeation. The above result also implies the intrinsic virtue of anti-trade off phenomenon in the MOF-in-COF membrane achieving remarkably improved separation selectivity without sacrificing too much the permeance by incorporation of only a tiny amount of MOFs in COF layer.

Some of the above discussion has been added to the revised manuscript and **Fig. R5** as **Supplementary Fig. 15** has been added to the Supplementary Information.

Fig. R5 Changes of gas permeance performance with the reaction time for ZIF-67 after addition of 2-meIM. (Equimolar binary gases at 298 K and 1 bar.)

Reviewer #3 (Remarks to the Author):

General comments:

Interesting concept and fabrication process. The separation performance is impressive and indeed relevant for challenging gas separations. The characterization is quite comprehensive and elucidates most of the structural features of interest. I would recommend publication in NCOMMS with the following suggestions for revision:

Response to general comments: Many thanks for the excellent comments and high affirmation of our research work. According to the reviewer's suggestion, we have added some more comments on the interfacial interaction related to the membrane permeation performance in the revised manuscript. Moreover, further discussion and analysis on the microstructure and interfaces in relationship with the separation performance have been added by combining the Maxwell predictions and the comparison with the experimental permeation results. We prepared pure ZIF-67 membrane and compared its gas permeation performance with the MOF-in-COF membrane. In addition, we have cited the paper mentioned by the reviewer and more other relevant publications to the reference. Also some comments in connection to the ZIF-zeolite case have been added in the revised manuscript. Please see the detailed point-by-point responses as follows:

Comment 1: There could be more discussion on the influence of the MOF-COF interfaces. This is clearly important considering the 1D nature of the COF pores: an unfavorable connection between the MOF and COF could completely shut down permeation. I understand it is extremely difficult to characterize it in detail, but some additional comments are desirable. The authors discuss partial pore blocking, can this be connected to the interfacial structure?

Response 1: This is a good suggestion. Yes, if the connection between the MOF and COF is unfavorable, it is very possible to completely block the 1D channel of COF and thereby lead to the shutdown of gas permeation. But in this study, after the confined growth of MOFs the MOF-in-COF membrane can still remain an ultrahigh H₂ permeation of ~ 3200 GPU as compared to that (~ 4800 GPU) of the pristine COF membrane. This implies that in the confined microscopic space, interfacial interaction (Co-N) between the MOFs and COFs could appropriately fix the MOF crystal plane orientation which was able to allow the fast molecule diffusion through the six-membered ring windows of MOF (such as ZIF-67) during gas permeation. Moreover, due to the confinement synthesis and existence of this interfacial interaction, the MOF formed in the 1D channel of COF could have smaller effective pore size than that of the bulk MOF, probably as a result of the greater rigidity of the confined MOF lattice^[R4,R5]. This means the effective pore aperture of the ZIF-67 formed inside the COFs is much less than ~ 0.4 nm (after ligand flipping motion) and close to the calculated window size of ~ 0.34 nm, which could be explained from the low permeance of other bulky gases (such as CO₂, CH₄) and the excellent sieving performance. For example, in comparison with the pristine TpPa-1 membrane, the CH₄ permeance of the ZIF-67-in-TpPa-1 membrane was decreased by 82 % (from 527 GPU to 105 GPU), but the H₂/CH₄ selectivity of the latter can reach 33.3, far higher than that of the pristine TpPa-1 membrane (~ 9) and even the as-synthesized ZIF-67 membrane (~ 13, please see the detailed performance data in **Response 4**) under the same measurement conditions.

Of course, incorporation of MOF would take up some space of the COF channels and surely impede molecular

transfer, to a certain extent. Moreover, during such confined growth, probably it could lead to the formation of few amorphous MOFs blocking the COF pores. Indeed, the interfacial structure is unknown and also extremely difficult to be characterized. But from the superior gas permeation result, it suggests a desired connection between the COFs and MOFs that allows access to the defined and ultrafast transfer pathways for H₂ molecules through the MOF-in-COF layer. In addition, in this study, due to that the MOF is most probably grown close to the orifice of 1D channels of COF, not in the entire COF layer, and therefore the diffusion resistance is relatively small. This is another main reason for the high gas permeation of the MOF-in-COF membrane. According to the reviewer's suggestion, we have added some comments related to the interfaces between MOF and COF in the revised manuscript.

Added references in the revised manuscript (numbered as Ref [61], Ref [62])

[R4] Zhou, S., Wei, Y., Li, L., Duan, Y., Hou, Q., Zhang, L., Ding, L. X., Xue, J., Wang, H. & Caro, J. Paralyzed Membrane: Current-Driven Synthesis of a Metal-Organic Framework with Sharpened Propene/Propane Separation. *Sci. Adv.* **4**, eaau1393 (2018).

[R5] Hou, Q., Wu, Y., Zhou, S., Wei, Y., Caro, J. & Wang, H. Ultra-Tuning of the Aperture Size in Stiffened ZIF-8_Cm Frameworks with Mixed-Linker Strategy for Enhanced CO₂/CH₄ Separation. *Angew. Chem. Int. Edit.* **58**, 327-331 (2019).

Comment 2: What is the estimated "loading" (in vol or wt%) of the COF sheets in the MOF matrix in the final membranes? Any comment on how this can be varied?

Response 2: The MOF volume fraction (vol%) could be obtained based on the Co/O atomic (molar) ratio measured by XPS on the MOF-in-COF selective layer. According to the chemical formula of ZIF-67 (Co(2-MeIm)₂) and TpPa-1 (C₁₈H₁₂N₃O₃), and their densities (0.903 and ~ 0.6 g/cm³, respectively), the estimated loading of ZIF-67 in the MOF-in-COF matrix is about 13.3 vol%. In this study, the MOF loading could be varied by controlling the precursor concentration and the synthesis time during the two-stage immersion process for the confined growth of ZIF-67.

Comment 3: Also, it is very simple to classically estimate the performance of a membrane containing sheets/flakes (Cussler-type models). In the limit of high aspect ratio flakes, it reduces essentially to resistance in series model. Disagreement of the experimental separation results with Cussler-type model is usually a good sign that interface effects are in play.

Response 3: Thanks for the reviewer's suggestion. Considering the near-spherical shape of the MOF dispersed in the COF layer, the Cussler-type models^[R6,R7] mentioned are rarely appropriate for the permeation performance prediction of the MOF-in-COF membrane. Therefore, in this study we could simply estimate and quantitatively analyze the separation performance for H₂/CH₄ mixture by the Maxwell model (equation R1)^[RS-R10].

$$P_{MOF-in-COF} = P_{COF} \frac{2(1 - \theta) + \phi(1 + 2\theta)}{(2 + \theta) + \phi(1 - \theta)} \quad \text{equation R1}$$

where $P_{MOF-in-COF}$ is the MOF-in-COF membrane gas permeance; θ is the volume fraction of the MOF; and Φ is the permeance ratio of the two phases, P_{MOF}/P_{COF} , which could be obtained by the adsorption and gas permeation data.

The calculated $P(H_2)_{MOF-in-COF}$ is about 4014 GPU higher than that of the experimental value of 3252 GPU. Simultaneously, the predicted H_2/CH_4 selectivity is about 9.2, much lower than the measured value of 33.3. The predicted result is indeed not consistent with the experimental separation result. This reveals that the formed MOFs are not simply dispersed in the COF matrix, but with a good interfacial interaction between the MOFs and COFs. Moreover, the smaller predicted selectivity further suggests the formation of MOF-in-COF pore structure in the resulting membrane which endows the precise molecular sieving and ultrafast transfer channels for the targeted molecules during the gas permeation process. The above discussion and analysis have been added into the revised manuscript.

Reference (R[8] was added and re-numbered as Ref[63] in the revised manuscript)

[R6] Yang, C. F., Smyrl, W. H. & Cussler, E. L. Flake alignment in composite coatings. *J. Membr. Sci.* 231, 1-2 (2004).

[R7] Cussler, E. L. Membranes containing selective flakes. *J. Membr. Sci.* **52**, 275-288 (1990).

[R8] Vinh-Thang, H. & Kaliaguine, S. Predictive Models for Mixed-Matrix Membrane Performance: A Review. *Chem. Rev.* **113**, 4980-5028 (2013).

[R9] Bouma, R. H. B., Checchetti, A., Chidichimo, G. & Drioli, E. Permeation through a Heterogeneous Membrane: The Effect of the Dispersed Phase. *J. Membr. Sci.* **128**, 141-149 (1997).

[R10] Maxwell, J. C. A Treatise on Electricity and Magnetism; Clarendon Press: Oxford, 1873.

Comment 4: The authors compare the COF-MOF membrane to a pure COF membrane, but they could also analyze the COF-MOF membrane in relation to a pure MOF membrane? Is the performance of a pure ZIF-67 membrane for H_2/CO_2 and H_2 /hydrocarbon separations known (or predicted from simulations)? A brief discussion on the advantage of spacing the ZIF-67 regions with COF sheets could be given.

Response 4: To further highlight the performance and structure advantages of the MOF-in-COF membrane, we prepared the pure ZIF-67 membrane via the seeded-assisted *in-situ* growth approach, and also tested the gas permeation performance. The specific preparation process is shown as follows:

First, ZIF-67 nanoseeds were prepared at room temperature. In a typical synthesis, 0.546g of $Co(NO_3)_2 \cdot 6H_2O$ and 0.616 g of 2-meIM were dissolved in a 15 mL of methanol, respectively. After blending of the two methanolic solution and sonication for 15 min, the dark-purple precipitate was collected by centrifugation, washed by methanol for several cycles. The obtained ZIF-67 nanoseeds colloid without drying was directly dispersed in 0.1 g PEI (50 wt% in water) aqueous solution (4 mL) in presence of 10 mg $NaHCO_3$, which was then treated under ultrasonic conditions for 10 min. Afterwards, an activated $\alpha-Al_2O_3$ disk substrate was dip-coated in the above seeding solution for 20 s, and dried in air. If needed, this procedure could be repeated to ensure a satisfied coverage of ZIF-67 seeds. For the synthesis of ZIF-67 membrane, the seeded $\alpha-Al_2O_3$ disk was placed vertically in a 50 mL Teflon-lined stainless steel vessel, which was filled with synthesis solution (0.11 g of $Co(NO_3)_2 \cdot 6H_2O$ and 2.27

g of 2-meIM in a mixed solvent of 2.5 ml of methanol and 17.5 ml of D.I. water). The autoclave was kept in an oven at 120 °C for 48 h. After natural cooling, the as-prepared membrane was washed with methanol and dried at 80 °C overnight.

As shown in **Fig. R6**, the membrane surface consists of continuous and intergrown ZIF-67 crystals, no cracks, pinholes, or other defects are visible. The thickness of MOF layer is about 1 μm , nearly the same as the thickness of the MOF-in-COF membrane. Moreover, a typical XRD pattern of ZIF-67 layer (**Fig. R7**) shows the purity and high degree of the crystallinity, because all peaks match well with those of simulated ZIF-67.

Gas permeation performance of the ZIF-67 membrane was measured by separating an equimolar H_2/CO_2 , H_2/CH_4 mixtures, respectively, at room temperature (298 K) and 1 bar. As shown in **Fig. R8**, the ZIF-67 membrane has separation factors of about 18, and 13 for H_2/CO_2 , and H_2/CH_4 gas pairs, respectively, and simultaneously the H_2 permeance is about 440 GPU. Both selectivity and H_2 permeance are far below the MOF-in-COF membrane. The relatively low separation selectivities are as expected because there are void/low-density or nonselective regions easily existed in the MOF layer during the intergrowth process by the seeded-assisted in-situ solvothermal approach. Moreover, the gate opening flexibility in ZIF-67 would also result in the poor molecular sieve performance. The low permeance is mainly due to the intrinsic ultramicropore system of ZIF-67 and perhaps as a sequence of the formation of impermeable regions caused by the intergrowth of crystals.

As has been stated in **Response 1**, incorporation of MOF into the COF layer could probably lead to the rigidity of the confined MOF lattice, and in this case, with narrower pore size would obtain more effective sieving performance for H_2 separation from other bulky gases. Moreover, in this study because the MOF is most possibly grown close to the orifice zone of the 1D channels of COF (not zigzag paths), not in the entire COF layer, there is a smaller diffusion resistance in the MOF-in-COF layer. In a word, spacing the MOF in the COF enables the formation of MOF-in-COF micro/nanopore network, which endows the precise molecular sieving and ultrafast transfer channels for penetrating molecules through the membrane. Therefore, with the synergy of this two different nanoporous materials, the MOF-in-COF membrane could exhibit both an ultrahigh H_2 permeance and a remarkable enhancement in separation selectivity for gas mixtures as compared to individual COF and MOF membranes. According to the reviewer's suggestion, we have added more discussion on the advantages of MOF-in-COF membrane structure in the revised manuscript. **Fig. R6-R8** have been added into the Supplementary Information as **Supplementary Fig. 12-14**.

Fig. R6 (a) Surface and (b) cross-sectional SEM images of as-synthesized ZIF-67 membrane. Insert shows the

optical photograph of membrane.

Fig. R7 XRD patterns of as-synthesized ZIF-67 membrane and simulated ZIF-67.

Fig. R8 Permeance and separation factors of the ZIF-67 membrane for equimolar binary gases at 298 K and 1 bar.

Comment 5: It may be worth mentioning recent publications on MOF-zeolite nanosheet membranes (e.g., Eum et al, ACS Appl Mater Interf, 2020) as a similar concept. The above paper also states questions regarding the interfaces and MOF crystallization. In the present context, is the ZIF-67 crystallized between the COF sheets the same as bulk ZIF-67 (e.g., rigidity of the framework)? These questions do not need to be answered in this paper, but stating them in connection to the ZIF-zeolite case would be useful for the community.

Response 5: Many thanks for the reviewer's suggestion. We have carefully read this excellent paper (e.g., Eum et al, ACS Appl Mater Interf, 2020) recommended by the reviewer. In this publication, the authors stated that "the crystallization of the ZIF-8 matrix in the confined microscopic spaces between the densely packed MFI nanoparticles/nanosheets may lead to the formation of crystalline or even partially amorphous regions with considerably smaller effective ZIF-8 pore size than that of bulk ZIF-8 (perhaps as a consequence of the greater rigidity of the confined ZIF-8 lattice)." Moreover, the structure of the nanoscopic interfaces between the ZIF-8 and MFI regions may have significantly different selectivity than bulk ZIF-8. We are in complete agreement with this point. Similarly, for the confined synthesis of MOF inside COF layer, we also believe that probably the ZIF-67 framework would become more rigid, and have narrower effective pore size than that of the bulk MOF. In addition, there is a possibility of formation of amorphous MOF region inside the COF layer, and also the interfaces between these two phases could play a role in the gas permeation and separation process. Otherwise, there could be no

obvious synergistic performance enhancement in the MOF-in-COF layer.

According to the reviewer's suggestion, we have cited this paper and stated the above comments in connection to the ZIF-zeolite case in the revised manuscript. Moreover, we also added another paper (e.g., Rashidi, *Angewandte Chemie*, 2019, 131(1): 242-245) related to the all-nanoporous hybrid membranes to the reference.

Added references (re-numbered as Ref [64], Ref [65] in the revised manuscript)

[R11] Eum, K., Yang, S., Min, B., Ma, C., Drese, J. H., Tamhankar, Y. & Nair, S. All-Nanoporous Hybrid Membranes: Incorporating Zeolite Nanoparticles and Nanosheets with Zeolitic Imidazolate Framework Matrices. *ACS Appl. Mater. Interfaces*. **12**, 27368-27377 (2020).

[R12] Rashidi, F., Leisen, J., Kim, S. J., Rownaghi, A. A., Jones, C. W. & Nair, S. All-Nanoporous Hybrid Membranes: Redefining Upper Limits on Molecular Separation Properties. *Angew. Chem. Int. Edit.* **131**, 242-245 (2019).

REVIEWER COMMENTS

Reviewer #1 (Remarks to the Author):

Accept

Reviewer #2 (Remarks to the Author):

As claimed by the authors, MOF-in-COF is the key concept of this work and of course this technique will provide a promising method to fine tune the molecular sieving performance of a porous material, especially when it was fabricated as a membrane. However, the revised version failed to provide the clear direct proofs to support this new concept.

1. The authors claimed that the diameter of TpPa-1 is 1.83 nm. But the pore size distribution indicated that the pore width of TpPa-1 is ca. 1.2 nm (Fig 2e). The COF and MOF are not so match.
2. The pore size distribution analysis (Fig 2e) showed that TpPa-1 had some mesopores ca. 2-4 nm or larger. That could be the defects or the void from crystal-packing. Such void or defects will provide the space large enough for ZIF-67. The discussion of this part is omitted by the authors.
3. XRD patterns of MOF-in-COF powders scraped from the substrate were too rough to identify which peaks belonged to ZIF-67. Even though some "peaks" can be attributed to the ZIF-67, it cannot exclude the possibility of the ZIF-67 grown mainly in the defects.
4. If the ZIF-67 fixed up the defects and some coordination compounds formed inside the COFs (only occupy part of the pores), the overall performance of such membrane will be "promoted".
5. The content of "ZIF-67" synthesized in the COF pores was not too low to be characterized. But they are mainly amorphous species.

In conclusion, I cannot recommend its publication in Nature Communications.

Reviewer #3 (Remarks to the Author):

The authors have done an excellent and thorough job of responding to all my comments and making substantive revisions. The discussion added at different locations on the interface effects and the final details of the composite structure, is important and points to many possible research directions for the community. I also looked at the responses to the other reviewers' comments and found them to be similarly well done. At this point I am completely satisfied with this fine work and strongly recommend its publication in NCOMMS.

Response to the reviewers:
MOF-in-COF molecular sieving membrane for selective hydrogen separation

Hongwei Fan, Manhua Peng, Ina Strauss, Alexander Mundstock, Hong Meng & Jürgen Caro

Reviewer #1 (Remarks to the Author):

General comments:

Accept

Response: Thanks so much again for the reviewer's valuable suggestions and affirmation of our revised manuscript.

Reviewer #2 (Remarks to the Author):

General comments:

As claimed by the authors, MOF-in-COF is the key concept of this work and of course this technique will provide a promising method to fine tune the molecular sieving performance of a porous material, especially when it was fabricated as a membrane. However, the revised version failed to provide the clear direct proofs to support this new concept.

Response to general comments: Many thanks for the reviewer's comments and high praise of our research work. We really understand the reviewer is most concerned with the growth of ZIFs in COFs or in the defect interfaces. It is worth emphasizing that broadly speaking, **whether the ZIFs grew inside the COFs or in the interfacial defects, are consistent with our MOF-in-COF concept.** Moreover, we believe most likely, the MOF grew both inside the COFs and in the potential defects in COF layer. On one hand, the pore channel of COFs is large enough to accommodate the growth of ZIF-67, even the growth of amorphous ZIF-67. On the other hand, the XRD patterns combined with other supporting evidence such as XPS and adsorption-desorption isotherms as well as the superior membrane performance could confirm the ZIF-67 synthesized inside the COF layer.

In addition, there is no doubt that the intrinsic pore aperture of TpPa-1 is 1.83 nm in diameter. The experimental pore size distributions in Fig. 2e are indeed slightly smaller than the intrinsic pore size of TpPa-1. This is possibly due to the partly occupied pores induced by staggered stacking of the 2D COF layers along the c direction, similar to many other reported results in the references. According to the reviewer's suggestion, we have added further discussions and proper interpretation for the ZIF-67 growth in the revised manuscript. Please see the detailed point-by-point response as below

Comment 1: The authors claimed that the diameter of TpPa-1 is 1.83 nm. But the pore size distribution indicated that the pore width of TpPa-1 is ca. 1.2 nm (Fig 2e).

The COF and MOF are not so match.

Response 1: Thanks for the reviewer's comments. TpPa-1 was first reported in 2012 by Banerjee and co-workers^[R1]. It shows a pore diameter of about 1.83 nm, based on the simulated patterns obtained using the eclipsed stacking model^[R1-R5]. So, there is no doubt that the intrinsic pore aperture of TpPa-1 is 1.83 nm in diameter. Therefore, the regular pore channel of TpPa-1 is large enough to accommodate the unit cell-sized MOFs.

In this study, the pore size distribution was calculated on the basis of nonlocal density functional theory model (NLDFE), which is a commonly used model for the analysis of pore size of COF materials. From Fig. 2e, the TpPa-1 has narrow pore size distributions of 1.1-1.7 nm, with peak maxima at about 1.3 nm. Indeed, the measured results of pore size distribution from BET are slightly smaller than the intrinsic pore size of TpPa-1 (~ 1.83 nm). This is possibly due to the partly occupied pores induced by staggered stacking of the 2D COF layers along the c direction. This phenomenon is very common for the 2D COFs, especially for the 2D ketoenamine-linked TpPa-1. **Fig. R2a-i** shows the statistical pore size distributions of the TpPa-1 reported in references^[R1, R3-R10]. It can be seen that some TpPa-1 also has a smaller pore size than the calculated value of 1.83 nm. Moreover, similar phenomenon occurred in the imine-based COF-LZU1 (**Fig. R2j** and **k**)^[R11]. In addition, we re-measured the pore distribution of TpPa-1 synthesized by using other approach, and also obtained a relatively small pore size compared to the theoretical value (1.83 nm) of TpPa-1, as shown in **Fig. R3**.

In brief, it is normal that there is a certain deviation between the theoretical value and experimental value. We have added some explanations for the pore size distribution of TpPa-1 in the revised manuscript. Even so, as can be seen from **Fig. 2e**, the experimental pore size distribution of ZIF-67-in-TpPa-1 is concentrated in the range of 0.29-0.5 nm, significantly less than the original TpPa-1. This tendency indicates the

pore space was occupied after incorporation of MOFs, which effectively narrowed the COF pore size in a suitable range for gas permeation studies.

Fig. R1 (a) PXRD patterns of the observed (red) and simulated (black) TpPa-1 with insets illustrating the chemical structure^[R1].

Fig. R2 Statistical pore size distributions of (a-i) TpPa-1 and (j) 2D imine-based COF-LZU1 with corresponding chemical structure (k).

Fig. R3 Nitrogen adsorption-desorption isotherms with inserted pore-size distribution of TpPa-1 measured at 77 K (Adsorption, closed; desorption, open).

Reference

- [R1] Kandambeth, S., Mallick, A., Lukose, B., Mane, M. V., Heine, T. & Banerjee, R. Construction of crystalline 2D covalent organic frameworks with remarkable chemical (acid/base) stability via a combined reversible and irreversible route. *J. Am. Chem. Soc.* **134**, 19524-19527 (2012).
- [R2] Chandra, S., Kandambeth, S., Biswal, B. P., Lukose, B., Kunjir, S. M., Chaudhary, M., Babarao, R., Heine, T. & Banerjee, R. Chemically Stable Multilayered Covalent Organic Nanosheets from Covalent Organic Frameworks via Mechanical Delamination. *J. Am. Chem. Soc.* **135**, 17853-17861 (2013).
- [R3] Biswal, B. P., Kandambeth, S., Chandra, S., Shinde, D. B., Bera, S., Karak, S., Garai, B., Kharul, U. K. & Banerjee, R. Pore surface engineering in porous, chemically stable covalent organic frameworks for water adsorption. *J. Mater. Chem.*

A, **3**, 23664-23669 (2015).

[R4] Pachfule, P., Panda, M. K., Kandambeth, S., Shivaprasad, S. M., Dáz, D. D. & Banerjee, R. Multifunctional and robust covalent organic framework–nanoparticle hybrids. *J. Mater. Chem. A* **2**, 7944-7952 (2014).

[R5] Singh, V., Jang, S., Vishwakarma, N. K. & Kim, D. P. Intensified synthesis and post-synthetic modification of covalent organic frameworks using a continuous flow of microdroplets technique. *NPG Asia. Mater.* **10**, e456-e456 (2018).

[R6] Biswal, B. P., Kunjattu, S. H., Kaur, T., Banerjee, R. & Kharul, U. K. Transforming covalent organic framework into thin-film composite membranes for hydrocarbon recovery. *Sep. Sci. Technol.* **53**, 1752-1759 (2018).

[R7] Sheng, J. L., Dong, H., Meng, X. B., Tang, H. L., Yao, Y. H., Liu, D. Q., Bai, L. L., Zhang, F. M., Wei, J. Z. & Sun, X. J. Effect of Different Functional Groups on Photocatalytic Hydrogen Evolution in Covalent–Organic Frameworks. *ChemCatChem* **11**, 2313-2319 (2019).

[R8] Ghosh, S., Khan, T. S., Ghosh, A., Chowdhury, A. H., Haider, M. A., Khan, A. & Islam, S. M. Utility of Silver Nanoparticles Embedded Covalent Organic Frameworks as Recyclable Catalysts for the Sustainable Synthesis of Cyclic Carbamates and 2-Oxazolidinones via Atmospheric Cyclizative CO₂ Capture. *ACS Sustain. Chem. Eng.* **8**, 5495-5513 (2020).

[R9] Thote, J. T., Barike Aiyappa, H., Rahul Kumar, R., Kandambeth, S., Biswal, B. P., Shinde, D. B., Roy, N. C. & Banerjee, R. Constructing covalent organic frameworks in water via dynamic covalent bonding. *IUCrJ* **3**, 402-407 (2016).

[R10] Wei, H., Chai, S., Hu, N., Yang, Z., Wei, L. & Wang, L. The microwave-assisted solvothermal synthesis of a crystalline two-dimensional covalent organic framework with high CO₂ capacity. *Chem. Commun.* **51**, 12178-12181 (2015).

[R11] Ding, S. Y., Gao, J., Wang, Q., Zhang, Y, Song, W. G., Su, C. Y. & Wang, W. Construction of covalent organic framework for catalysis: Pd/COF-LZU1 in Suzuki–Miyaura coupling reaction. *J. Am. Chem. Soc.* **133**, 19816-19822 (2011).

Comment 2: The pore size distribution analysis (Fig 2e) showed that TpPa-1 had some mesopores ca. 2-4 nm or larger. That could be the defects or the void from crystal-packing. Such void or defects will provide the space large enough for ZIF-67. The discussion of this part is omitted by the authors.

Response 2: According to the reviewer’s suggestion, we have added further discussions about the ZIF-67 growth in the defects or voids in the revised manuscript. Please see the added part as below.

“In addition, there exist possibly some defects at nanoscale (a few nm or even larger) at the COF grain boundaries or in the gaps between the COF layers. Such defective voids provide also sufficient space for ZIF-67 formation. Some MOFs, therefore, might also grow in the COFs grain boundaries (Fig. 2f) thus repairing these defects, which improves gas selectivity.”

Comment 3: XRD patterns of MOF-in-COF powders scraped from the substrate were too rough to identify which peaks belonged to ZIF-67. Even though some “peaks” can be attributed to the ZIF-67, it cannot exclude the possibility of the ZIF-67 grown mainly in the defects.

Response 3: Many thanks for the reviewer’s comments. We did not rule out the possibility of ZIF-67 grown in the defects. Instead, we have already mentioned and discussed this case on page 10, line 17-21, in our previous version and revised manuscript. Despite the weak ZIF-67 diffraction peaks, also it cannot exclude the possibility of ZIF-67 growth inside the COFs. Considering the large enough pore size of TpPa-1 (please see **Response 1**), we believe the ZIF-67 could grow inside the 1D channel of COFs.

Moreover, it is certain that the number of the potential defects in COF selective layer is far less than that of the COF pores. Otherwise, the supported TpPa-1 layer without ZIF-67 (or pristine TpPa-1 membrane) will show extremely low or even no separation selectivity for gas mixtures. However, in this study the as-synthesized TpPa-1 layer still has a certain separation selectivity for equimolar gas pairs of H₂/CO₂ (11.9), H₂/CH₄ (9.3), H₂/C₃H₆ (45.8), and C₃H₈ (55.6), exceeding the corresponding Knudsen constants (4.7, 2.8, 4.6, 4.7). This clearly indicates only a small number of defects in the COF layer. In this case, most ZIF-67 was formed inside the COF pore channels.

Even so, broadly speaking, **whether the MOF grew inside the COF pore channels or in the defects, this all is consistent with our proposed MOF-in-COF concept of “confined growth of metal organic framework (MOFs) inside a supported COF layer to prepare MOF-in-COF membranes” in the manuscript.**

Comment 4: If the ZIF-67 fixed up the defects and some coordination compounds formed inside the COFs (only occupy part of the pores), the overall performance of such membrane will be “promoted”.

Response 4: Thanks for the reviewer’s comments. We agree that “if the ZIF-67 fixed up the defects and some coordination compounds formed inside the COFs (only occupy part of the pores), the overall performance of such membrane will be “promoted””. But we believe this is an extreme case, and it is unlikely that the ZIF-67 only existed in the defects and some coordination compounds formed inside the COFs (only occupy part of the pores). Because the pore channel of COFs is large enough and also suitable for growth of ZIF-67, even the growth of amorphous ZIF-67. Most likely, the MOF grew both inside the COFs and in the potential defects in COF layer.

Moreover, as we mentioned in **Response 3**, even the growth of ZIF-67 in the defects or the growth of amorphous ZIF-67 inside the COFs, it still fits our proposed concept of “MOF-in-COF membrane”. Furthermore, this strategy has been verified to be effective to significantly improve the overall performance of the COF membranes for gas separation.

Comment 5: The content of “ZIF-67” synthesized in the COF pores was not too low to be characterized. But they are mainly amorphous species.

In conclusion, I cannot recommend its publication in Nature Communications.

Response 5: Many thanks for the reviewer’s comments. Indeed, the existence of amorphous specie is one of the possible reasons. Besides, the tiny dimension of ZIF-67 with a highly dispersed state (or isolated by the COF matrix) in the COF layer is also an important reason not to be ignored. So it is really not easy to directly characterize the specific shape of the MOF inside the COFs. Despite this, the XRD patterns combined with other supporting evidence such as XPS and adsorption-desorption isotherms as well as the superior membrane performance could confirm the ZIF-67 synthesized inside the COF layer. Moreover, as we mentioned in our previous response, we are working on the growth of MOF inside a larger-pore 2D COFs (~ 5 nm) to synthesize new MOF-in-COF membranes. The specific structural features of the MOFs formed in the confined 1D channel of 2D COF and growth mechanism will be systematically elucidated in our next research work.

We appreciate the reviewer’s inspirational suggestions, and added proper interpretation for this part of MOF characterizations in the revised manuscript. In a word, we are expecting that our “MOF-in-COF” strategy could bring a new avenue for constructing next-generation molecular sieving membrane, pore surface engineering in COFs and other promising COF-based functional materials with great application potential.

Reviewer #3 (Remarks to the Author):

General comments:

The authors have done an excellent and thorough job of responding to all my comments and making substantive revisions. The discussion added at different locations on the interface effects and the final details of the composite structure, is important and points to many possible research directions for the community. I also looked at the responses to the other reviewers' comments and found them to be similarly well done. At this point I am completely satisfied with this fine work and strongly recommend its publication in NCOMMS.

Response: Thank the reviewer very much for the positive remarks, kind encouragement and valuable guidance on our manuscript.